# Scale-breaks of suspended sediment rating in large rivers in Germany induced by organic matter

Thomas O. Hoffmann[1,2], Yannik Baulig[1], Helmut Fischer[1], Jan Blöthe[2]

[1]Bundesanstalt für Gewässerkunde, 56068 Koblenz, Germany
[2]Department of Geography, University of Bonn, 53115 Bonn, Germany

*Correspondence to*: Thomas O. Hoffmann (thomas.hoffmann@bafg.de)

**Abstract.** Understanding the transport of suspended sediment and associated nutrients is of major relevance for sustainable sediment management aiming to achieve healthy river systems. Sediment rating curves are frequently used to analyze the suspended sediments and their potential sources and sinks. Here we are using more than 750 000 measurements of suspended sediment concentrations (*SSC*) and discharge (Q) collected at 62 gauging stations along 19 waterways in Germany based on the suspended sediment monitoring network of the German water and shipping authority, which started in the 1960s. Furthermore, we analyse more than 2000 measurements of the loss on ignition (*LOI*) of suspended matter at two stations along the rivers Moselle and Rhine to provide a proxy of the relative contributions of mineral load and organic matter. SSC and LOI are analysed in terms of the power law rating curve to identify discharge dependent controls of suspended matter.

Our results indicate that for most studied gauging stations, rating coefficients are not constant over the full discharge range, but there is a distinct break in the sediment rating curve, with specific *SSC-Q* domains above and below this break. The transition of the rating exponent likely results from increased supply of mineral suspended sediments from hillslope erosion at high flow and a shift of the organic matter sources from aquatic biomass-derived organic matter (i.e. high %*LOI*) at low flow, to mineral-associated organic matter with low %*LOI* eroded from hillslopes at higher flow. Based on these findings we developed a conceptual rating model for large (> 10000 km$^2$) and low-turbidity (SSC < 1000 mg l$^{-1}$) rivers separating the mineral and organic fraction of the suspended matter in the Germany waterways. This model allows evaluating the sources of the mineral and organic fraction of the suspended matter and facilitates new insights into the first order control of discharge on the quality and quantity of suspended sediments.

## 1 Introduction

Suspended sediment dominates sediment transport of almost all lowland rivers of the world (Naden, 2010; Walling, 1996), and represents 90-95% of the global riverine sediment load to the coastal oceans (Syvitski et al., 2005). Silt and clay particles, which comprise the dominant grain size fraction of suspended sediments, form an important transport medium for

nutrients, pollutants and contaminants. Sustainable sediment management aiming to achieve healthy river systems therefore requires a sound understanding of the sources and sink of suspended sediment along the riverine flow paths.

Transport of suspended sediment is strongly conditioned by sediment characteristics (Owens et al., 2005; van Rijn, 1984; Walling et al., 2000). Size and density of sediment particles control their propensity to settle within the turbulent flow of the river, counteracting gravitational settling (Naden, 2010; Partheniades, 2009). Size and density of fine suspended particles in turn affects their affinity to form aggregates and flocs, due to strong cohesive forces between fine grain particles (Winterwerp and Van Kesteren, 2004). Depending on sediment sources, suspended particles are either mineral, organic, or a

combination of both. Erosion of (organic-rich) topsoil from either hillslopes or floodplains represents an important source of suspended sediment (mainly silt and clay) and supplies large amounts of (allochthonous) organic matter with site-characteristic carbon contents (Hoffmann et al., 2009). Sediment supply generated by surface runoff in response to intensive and/or long-lasting rainfall events typically results in increased levels of suspended sediment concentration (SSC) in river channels during higher discharges (e.g. Asselmann, 2000; Gray, 2018).

In addition to allochthonous suspended matter, phytoplankton is an important source of organic suspended matter that is autochthonously produced within rivers. Especially during spring and summer months, when phytoplankton growth is supported by high water temperatures, sufficient light, and high nutrient levels, autochthonous organic matter may dominate the total suspended load in many large lowland rivers and those with intense agricultural land use within the river catchment (Cloern, 1999; Hillebrand et al., 2018; Thorp and Delong, 2002). Water flow velocities regulate the water residence times,

which in turn affect the time available for phytoplankton growth in river systems. Low flow conditions with increased residence times provide favourable conditions for phytoplankton biomass accumulation including algal blooms with high organic SSC. In contrast, short residence times can strongly reduce the share of autochthonous biomass in suspended sediments because phytoplankton growth rates cannot compensate the downstream transport at higher discharges, even if temperature, light availability, and nutrient levels are not limiting phytoplankton growth (Fischer, 2015; Quiel et al., 2011).

Additionally, at high flow runoff and erosion supply materials from outside the channel that swamp the within-river production. Thus, a negative relationship between autochthones organic matter load and discharge is observed in many river systems (Gomez et al., 2003; Goñi et al., 2014; Hilton et al., 2012; Moreira-Turcq et al., 2013), in contrast to allochthonous suspended matter.

Besides physical factors controlling the abundance of phytoplankton in river systems, several studies stress the importance of

biological controls. For instance, Hardenbicker et al. (2016) suggest that low phytoplankton concentrations in the Rhine are at least partly the result of losses due to grazing by the invasive bivalve mollusk *Corbicula fluminea*, which increased in density since the early 1990s, while phytoplankton declined during the same time. Furthermore, predicting the characteristics of the suspended matter is confounded by the heterogeneous and composite structure of flocs and aggregates that are composed of mineral particles as well as living and dead organic matter (Winterwerp et al., 2006). The size of the flocs is a

function of the turbulence-induced collision of suspended particles and the cohesive and adhesive forces between the flocs. The latter is strongly controlled by the grain size and the organic matter of the suspended particles. Their size and density, in

turn, affect the transport conditions, with large and dense flocs being predominantly deposited, while flocs with a high organic matter content and a low density are transported over long distances (Winterwerp et al., 2006).

Sediment rating curves are frequently used to analyse the transport conditions of suspended sediments and their potential
sources and sinks (Asselmann, 2000; Cohn et al., 1992) or to predict suspended sediment yields at ungauged or unfrequently gauged stations (Ferguson, 1986; Horowitz, 2003; Morehead et al., 2003; Syvitski et al., 2000). Rating curves plot $SSC$ as a function of water discharge $Q$. The temporal aggregation (or resolution) depends on the approach and available data and ranges from 15 minutes to annual averages. In many cases, there is a close link between both variables that is mostly described by a power law:


$$SSC = aQ^b, \hspace{6cm} (1)$$

where $a$ and $b$ are coefficients that depend on the characteristics of the river system. $a$ represents the SSC at unit discharge and the exponent $b$ has been discussed in terms of sediment availability and the erosivity of the stream (Asselmann, 2000;
Syvitski et al., 2000). While $a$ varies over several orders of magnitude, depending on the river system characteristics, values of $b$ are typically more confined and range between 0.2 and 2.0 (Syvitski et al., 2000), with lower values in arid environments (i.e., 0.2 to 0.7) and higher values in humid, temperate river systems (i.e. 1.4-2.5, based on Reid and Frostick, 1987). However, small changes in the rating exponent $b$ can cause large changes in SSC, which are in the same order of magnitude as the changes imposed by the (large) variability in $a$ (Syvitski et al., 2000). Using Eq. 1, many studies found a
strong negative relationship between $a$ and $b$ (Asselmann, 2000), which is however not a matter of the natural balance between the two rating parameters (as proposed by Syvitski et al., 2000) but an artefact of the statistical analysis as the units of $a$ are dependent on $b$. Warrick (2015) suggests using normalized $Q$ and $SSC$ values to avoid this confusion and provide a statistically sound rating analysis (see also method section).

In most cases, observed $Q$ and $SSC$ scatter strongly around the regression line from Eq. 1. Deviations from the simple power
law haven been shown to result from i) hysteresis effects during single flood events (Aich et al., 2014; Zuecco et al., 2016), ii) seasonal changes of water and sediment sources or flow hydraulics (Asselmann, 2000; Morehead et al., 2003) or iii) long-term trends of changing sediment supply (Warrick, 2015). Event-based deviations are associated to: i) clock-wise hysteresis (i.e. the SSC-peak precedes the $Q$-peak ) with a rapid SSC-increase, due to within-channel mobilization of suspended sediment and subsequent sediment exhaustion, or ii) anti-clock-wise hysteresis (i.e. maximum $Q$ precedes the $SSC$-peak),
due to the long transport distance of sediment sources that are located within the catchment (e.g. arable land on inclined hillslopes with increased soil erosion rates) (Asselmann, 2000), as well as combinations of both within one event (leading to a complex hysteresis pattern). While the general processes affecting a rating relation in specific cases are well known, it is difficult to predict the rating behaviour as a result of the many confounding processes and linkages.

Recently, we have learned more about the controlling factors of sediment rating, but so far the effect of organic material on SSC rating is not sufficiently understood. An alternative control on SSC - Q relationships may be the varied contribution of organic matter to river sediment loads. However, many studies that have investigated the composition and loading of organic matter are limited to a relatively narrow window (~year) of sample collection, and tend to focus on steep upland catchments (e.g. Goñi et al., 2014; Hilton et al., 2012; Smith et al., 2013). In contrast, studies with a large number of samples tend to focus on total suspended sediment without considering their mineral and organic components. In this respect suspended sediment is equivalent to *seston*, a term used in ecological sciences to describe the total particulate matter including living organisms, organic detritus and inorganic particles (Naden, 2010; Wetzel, 2001). Consequently, most sediment rating studies, which focus on prediction of total SSC levels in river systems based on water discharge or on hysteresis effects of total SSC during single flood events, lump organic and inorganic particles into sediment rating curves. To the authors' knowledge, there is no study that rigorously investigates the influence of the variable mineral and organic fractions in river systems on the rating of sediment.

Here we hypothesize that the mineral and organic fractions of SSC in large German rivers are controlled by different and independent processes reflected in specific rating coefficients. We test this hypothesis by i) analysing the scaling of total suspended sediment with discharge, before we ii) differentiate between the scaling behaviour of the mineral and organic fractions of the suspended sediment against discharge. Furthermore, we develop a conceptional sediment rating model for large and low-turbidity rivers considering the mineral and organic fraction of the suspended sediment transport. To perform this study, we used a rich dataset on suspended sediment in the German waterways and analysed more than 750 000 suspended sediment measurements.

## 2 Method

### 2.1 Study sites

In this study we explore discharge and suspended sediment measurements at 62 gauging stations along 19 waterways in Germany. The studied rivers comprise the Danube, Rhine, Ems, Weser, Elbe and Oder, including some larger tributaries (Tab. 1 for details and Fig. 1 for location). The gauging stations cover contributing areas from 2,076 to 159,555 km², with a median of 24,424 km². The topography of the river catchments includes the steep high mountain terrain of the European Alps (e.g. Alpine Rhine and Danube) as well as the mountainous regions with various geological settings in Central Europe and the flat terrain of Northern Germany, which is mainly composed of glacial and fluvial Quaternary deposits. The long-term average discharge of all stations ranges from 9 to 2289 m³/s (Tab. 1). The strong control of contributing area on discharge is clearly reflected by the higher specific discharges (i.e., discharge per contributing catchment area) of the rivers Rhine and Danube (Jochenstein station), which are characterized by strong discharge contributions from the Alps (Fig. 2). In contrast, stations in the Elbe and Oder catchments show much lower (specific) discharges at a given catchment area, due to

lower rainfalls in the more continental climate, compared to the rivers in West and Central Germany, which are fed by elevated precipitation of the more maritime climate.

## 2.2 Suspended sediment monitoring in German waterways

Suspended sediment in German waterways is monitored daily using instantaneous water samples taken manually by the Federal Waterways and Shipping Administration (Wasserstraßen- und Schifffahrtsverwaltung des Bundes, WSV) at ~70 sampling locations. SSC monitoring started in 1965 and has accumulated long-term records that cover >30 years for many stations. Here, we selected only those stations from the monitoring network that are not located at artificial channels (with different flow regimes) and that cover periods longer than 10 years (Tab. 1 and Fig. 1), resulting in a total of 62 stations. Periods with more than 10 years were chosen to have sufficient data for the statistical analysis. Data from Maxau station at the river Rhine and from some tributaries haven been formerly presented by Asselmann (2000) and Horowitz (2003) in terms of a rating analysis and by Frings et al. (2014) and Frings et al. (2019) in terms of sediment budget calculations.

At each monitoring site, 5-liter bucket water samples were taken once each work day (excluding weekends and legal holidays) roughly in the top 30 cm of the water surface. During floods, the sampling frequency was increased up to 3 samples per day, unless sampling was stopped due to safety reasons. If more than one sample per day was taken, we used the mean SSC of all samples of that day. Limiting the water sampling to the top 30 cm slightly underestimates the average SSC in a channel cross-section (and thus the suspended sediment load). However, we argue that the rating behaviour does not significantly change, compared to depth integrated measurements (Morin et al., 2018).

Water samples were filtered using commercial coffee filters, which were weighed before and after filtering (under constant climatic conditions in the lab with 20°C air temperature und 50% air moisture) to calculate the daily SSC [kg/m³] (Hillebrand 2013). The use of coffee filters is cost-efficient and facilitates measuring SSC at a large number (i.e. 70 samples per day at the national scale) and with sufficient quality. However, these filters do not have a well-defined pore diameter and a significant fraction of clay is lost. In general, suspended sediment mainly contains silt (approx. 75%) and only a small fraction of clay (mostly 10-20 %) and fine sand (mostly below 10%) (for a detailed particle size analysis of the suspended sediment of the river Rhine see Hillebrand and Frings, 2017). Thus, clays are expected to comprise less than 20% of the suspended sediment, which agrees with comparisons of the suspended loads estimated using cellulose acetate filters (with pore diameter of 0,45 µm) and coffee filters. The latter underestimate loads by approximately 20% (Hillebrand et al., 2015).

For each SSC monitoring station, discharge is either measured at the station or nearby, without major tributaries entering the river between the SSC station and the discharge station. Water level is typically measured each 15 min and discharge is calculated using a rating curve. In this study, we used daily average discharge, which is then related to the daily SSC samples.

As shown in Tab.1, long-term averages of SSC for all stations range between 10.7 and 51.6 mg/l, with an average of 25 mg/l. Long-term discharge weighted averages of SSC are somewhat higher, ranging between 11.8 and 84.4 mg/l with a mean of 36 mg/l. Higher discharge weighted SSC reflects higher SSC at high discharge, which results in higher weights of increased

Kommentiert [TH1]: Morin JGR ES 2017

SSC. Similar to other national monitoring systems (e.g. Diplas et al., 2008; Habersack and Haimann, 2010; Spreafico et al., 2005; Thollet et al., 2018), SSC values for most stations used in this study include both the mineral and organic material of suspended sediment. Loss on ignition (*LOI*) and chlorophyll a (*Chla*), have been monitored since 1997 at two sampling locations, located immediately upstream of the confluence of the rivers Moselle and Rhine in Koblenz. At both stations,

water samples of 2 to 5 litters were taken at a weekly interval (in contrast to the daily sampling of the stations for the suspended sediment monitoring), resulting in a total of 1033 and 1056 samples from the Rhine and the Moselle, respectively (until end of 2017). Similar to SSC water samples, sampling for *LOI* was limited to the upper 50cm of the water surface using a bucket water sampler.

To measure the *LOI* at both stations, the water samples were filtered using a glass fiber filter with a pore size of about 1 μm

(Whatman GF 6, GE Healthcare, Germany). The filter was weighted empty (after heating at 500°C for 1 hour to combust organic remains on the filter) and after filtration. Between filtration and weighting the full filter was dried at 105°C for 24 hours, to obtain the total suspended sediment $SSC_{tot}$ (including the mineral and organic components). The whole samples was heated at 500°C for 1 hour, with the aim to combust the organic fraction of the suspended matter and to measure the *LOI*. In our study *LOI* is given as the ratio of the mass of organic matter (the mass LOI) to the total suspended sediment mass

(ranging from 0 to 1). Here we use *LOI* as a proxy for the organic matter content of the suspended sediments, despite the challenges that are related to this method (i.e. different protocols regarding the temperature and combustion length result in various LOIs and combustion may originate not only from organic matter but as well from clay-bound-water and carbonate decomposition).

Based on the *LOI*, we segregated the mineral ($SSC_{mrl}$) and organic ($SSC_{org}$) fraction of the SSC: $SSC_{org} = LOI \times SSC_{tot}$ and

195 $SSC_{mrl} = (1 - LOI) \times SSC_{tot}$.

For both stations, *Chla* was analysed in parallel with the *LOI* samples. *Chla* was used as a proxy of phytoplankton biomass in the rivers Rhine and Moselle. *Chla* concentrations were determined using German Standard Methods (DEW, 2007). Briefly, phytoplankton was filtered on glass-fiber filters and pigments were extracted with hot ethanol. Chlorophyll concentrations were determined photometrically (DR 2800, Hach Lange, Germany). *Chla* concentration (given in μg/l) was

200 transferred to living phytoplankton biomass using a C:Chla-ratio of 40 and a particulate organic matter (POM) to particulate organic carbon (POC) ratio of 0.42 (Geider, 1987; Hardenbicker et al., 2014; Hillebrand et al., 2018). The applied ratios represent global average conditions of C in *Chla* and POC in POM and allow only a first order estimate of living phytoplankton biomass. Therefore, we did not used this value to calculate the contribution of living phytoplankton biomass to the $SSC_{org}$. However, we used the ratio as a plausibility check to give a first order estimate of the origin of the suspended

organic matter based on a comparison of seasonal changes.

## 2.3 Rating analysis

To analyse suspended sediment as a function of discharge, we calculate sediment rating curves following Eq. 1. The interpretation of the coefficients $a$ and $b$ in Eq. 1 is impeded by their interdependence as illustrated by units of $a$ that depend on the exponent $b$: with $SSC$ having the dimension $M/L^3$ (M and L represent the dimension mass and length, and $L^3$ is equal to volume) and $Q^b$ having dimension of $L^{3b}/T^b$ (were T represent the dimension of time), the units of a are given by $MT^bL^{-(1+3b)}$ (note the direct dependency on the exponent $b$). To avoid this complication and to facilitate the comparison of rating curves between various stations, $SSC$ and $Q$ values are normalized by the geometric means ($SSC_{GM}$ and $Q_{GM}$, respectively) computed for each station according to Warrick (2015):

$$SSC/SSC_{GM} = a\,(Q/Q_{GM})^b \tag{2}$$

In Eq. 2, $a$ and $b$ are dimensionless. The exponent $b$ can be linked to the response of $SSC$ to changing discharge, and $a$ represents the normalized $SSC$ at $Q_{GM}$. The normalization using Eq. 2 does not have any effect on the exponent $b$ (i.e. the slope of the regression line does not change), but changes the absolute value of $a$.

For most studied gauging stations included in this study, $a$ and $b$ are not constant over the full discharge range, but there is a distinct break in the sediment rating curve, with specific $SSC$-$Q_w$ domains above and below this break. To estimate the discharge at which this break occurs ($Q_{br}$), we used three approaches. The first approach is based on the locally weighted scatter smoothing (lowess) regression curve (compare red dotted line in Fig. 3), which was calculated using the gplot-package in R according to Cleveland (1981). We defined $Q_{br}$ to be located at the maximum curvature of the lowess regression curve. In the second approach, we used a sequence of $n$ equally log-spaced discharges ($Q_i/Q_{GM}$, with 1<i<n and constant width of $\Delta Q = 10^{0.025}$) between $Q_{min}/Q_{GM}$ and $Q_{max}/Q_{GM}$ and extracted for each $Q_i/Q_{GM}$ the corresponding $SSC_i/SSC_{GM}$ value of the lowess regression curve. For each $i$ ($1 < i < n$) we build two subsets i) the low flow subset with data pairs smaller than or equal to $Q_i/Q_{GM}$ and the high flow subset with discharge larger than $Q_i/Q_{GM}$. We than applied a piecewise non-linear least square (NLS) regression to both subsets, which were both forced through the data pair ($Q_i/Q_{GM}$, $SSC_i/SSC_{GM}$). As $i$ increases (from $Q_1/Q_{GM}$ to $Q_n/Q_{GM}$), the mean absolute error (MAE) of the NLS regression of the low flow subset increases (first slowly while the break-point is approached and then more rapidly as the breakpoint is exceeded) and the MAE of the high flow decreases in a similar fashion (first rapidly and then slowly). As $Q_i/Q_{GM}$ approaches the break-point $Q_{br}$, the MAEs of both NLS regressions are small and their sum is at minimum. Thus, $Q_{br}$ was set to the $Q_i/Q_{GM}$ with the minimum of the sum of the MAE. The third approach is similar to the dual regression of the low and high flow subsets as applied in the 2nd approach. However, the third approach does not use the SSC-values of the lowess curve but uses log-binned median $SSC/SSC_{GM}$ of equally spaced discharge bins at the log scale (compare yellow points in Fig. 3). The median $SSC/SSC_{GM}$ values and the mid-point of each $Q$-class was split into low flow and high flow subsets and used for the piecewise regression analysis to identify $Q_{br}$ at which the sum of the MAE of both subsets was minimized.

At extreme discharges, rating relationships tend to be strongly scattered due to the low density of $SSC - Q$ data pairs. To estimate the $Q_{br}$, we thus excluded measurements with $Q$ smaller than the 1% and larger than the 99% discharge percentile of each station.

After the identification of $Q_{br}$ for each station, the coefficients in Eq. 2 were estimated for the low flow regime (i.e. all measured SSC-Q data pairs with $Q < Q_{br}$) and the high flow regime (i.e. all measured SSC-Q data pairs with $Q > Q_{br}$) using log-linear and non-linear least square regression (see. Tab. 2). Coefficients for the low flow regime are denoted by $a_l$ and $b_l$ and for the high flow regime by $a_h$ and $b_h$. To estimate the confidence intervals and thus to test for significant differences between rating exponents for the low-flow and high-flow regimes, we used a bootstrapping approach with 1000 replications (resulting in normal distributions of $b_l$ and $b_h$ with 1000 estimates) and compared the distributions of $b_l$ and $b_h$ using a t-test with a 95% confidence level.

The rating relations for $LOI$, $SSC_{org}$ and $SSC_{min}$ of the two stations at the rivers Moselle and Rhine in Koblenz were analysed the same way (similar to Eq. 2) as the $SSC$ at the 62 stations from the suspended sediment monitoring network.

## 3 Results

### 3.1 Rating of the total suspended sediment

For 52 out of 62 stations, $SSC - Q$ rating curves show a distinct break in scaling relation (for examples see Fig. 3) with similar values for $Q_{br}$ estimated from three different approaches (Tab. 2). For the remaining 10 stations, no distinct breakpoint is detectable (Fig. 3). After visual inspection and removal of non-plausible break-points of each station, we calculated the average $Q_{br}$ for each station. In general, breakpoints of the $SSC - Q$ relation range between $0.8 < Q_{br} < 1.9$, with 50% of all values ranging between 0.9 and 1.3 (Fig. 4) (the mean $Q_{br}$ over all stations is 1.2), indicating that the breakpoints of many stations are slightly larger than the geometric mean discharge.

Rating exponents for the low flow regime ($b_l$) range between -0.75 and 1.15 and for the high flow regime ($b_h$) between -0.6 and 2.45. In general, the distribution of $b_l$ peaks close to the median $b_l = 0.14$ (see Fig. 5). $SSC$ decreases as a function of $Q$ (i.e. $b_l < 0$) at 19 stations and increases with $Q$ at 33 stations. $b_h$ is < 0 at 11 stations and > 0 at 51 stations, with a median $b_h = 0.83$ (Fig. 5). 23 stations are characterized by strong increases of SSC with under high flow conditions (i.e. $b_h > 1$).

Patterns of spatial distribution become apparent (Fig. 6) for the rating coefficients $b_l$ and $b_h$. Highest $b_h$-values (positive rating in the high flow regime) are found along the Rhine and its tributaries, the Danube and Upper Weser, while the rivers in northern lowland Germany (mainly the Ems, Elbe and Oder rivers) show low $b_h$-values. This control is highlighted in Fig. 7b, which plots $b_h$ with respect to the fraction of hillslopes steeper than 10% in the contributing catchment area. Catchments with a higher fraction of steep slopes generate higher $b_h$-values compared to the lowland rivers, indicating higher sediment supply in catchments with more extensive hillslopes with slope gradients > 10%. Furthermore, the majority of the stations at the rivers Elbe and Oder, which are characterized by low annual rainfall in the contributing catchment, plot below the

regression line. In contrast to $b_h$, $b_l$ does not show a clear spatial pattern nor any relationship to the fractions of steep catchment areas (Fig. 7a).

**3.2 Rating curves for the mineral and organic fraction**

As noted above, *LOI* and *Chla* were measured at the Moselle and the Rhine just upstream from their confluence in Koblenz based on weekly sampling from 1997-2017. Despite the lower sampling frequency (samples at the suspended sediment stations are taken on each working day), the shorter monitoring period (SSC-monitoring started in 1964; Tab. 1), and a slightly different lab protocol, the rating behaviour of total SSC for both stations is similar to the rating curves for the other SSC stations along the rivers Rhine and Moselle (Fig. 8a-d and Tab. 2): i) rating breaks occur at 0.96 and 0.91 of the normalized discharge ($Q/Q_{GM}$) for the Rhine and the Moselle, respectively, and ii) the rating exponents $b_l$ (0.29±0.20 and -0.03±0.07 for the rivers Rhine and Moselle, respectively) and $b_h$ (2.26±0.18 and 1.54±0.14 for the Rhine and Moselle, respectively) are similar to the other stations along the Rhine and the Moselle. However, the lower number of measurements at the LOI-stations (approx. 1000 at each of the two stations) resulted in larger standard deviations of the parameter estimates ($\Delta b_l$ and $\Delta b_h$) based on the bootstrap regression.

Results from the LOI measurements of both stations show a higher organic matter contents in the Moselle (mean *LOI* = 0.385) compared to the Rhine (mean 0.237). *LOI* negatively correlates with discharge at both stations (Fig. 8e+f). However, the relationship for the Moselle is much better constrained. High *LOI* values cluster during the summer months (April – September), while low *LOI* values are more prominent during winter months (Fig. 8e+f and Fig. 9). Based on the bootstrap regression, a single power-law ($LOI = a \times (Q/Q_{GM})^b$) was fitted to the *LOI* data, resulting in rating exponents $b$ of $-0.51 \pm 0.03$ and $-0.47 \pm 0.01$, and $a$-coefficients of 0.202±0.003 and 0.319±0.006 for the Rhine and the Moselle, respectively. Based on the total *SSC* and *LOI*, the mineral fraction of the suspended sediment ($SSC_{mrl} = (1 - LOI) * SSC_{tot}$) was calculated. $SSC_{mrl}$ increases with discharge for both stations (Fig. 8c+d). Yet the variability for any given discharge is large (ranging approximately an order of magnitude) and increases at lower discharges.

In contrast to the *LOI*, *Chla* does not show significant changes with discharge. Fig. 8g+h shows dominantly low Chla values for the Rhine and Moselle. Increased *Chla*-values are mainly limited to lower discharges ($Q/Q_{GM} < 1$). Higher *Chla* values occur only during moderate flows in spring and summer. *Chla*-values in the Rhine peak in April, and in May at the Moselle (Figs. 8 and 9).

**4 Discussion**

Sediment rating analysis is challenged by the large scatter of single *SSC* measurements, which frequently ranges one order of magnitude around the regression lines. The scatter arises from hysteresis effects during single floods, and from seasonal variations of discharge and suspended sediment supply as well as long-term changes. Regression lines calculated in this study represent the average conditions at the gauging stations during the monitoring period (covering at least 10 years and at

maximum 55 years, see Tab.1). Therefore, we ignore long-term changes of $Q$ and $SSC$, which could be caused by climate and land use change or by changes in river management. Long-term trends of $SSC$ likely involves declines in $SSC$ for many stations along the German waterways (see for instance Hillebrand et al. 2018). This change will likely have an impact on the $a$ coefficient (which represents the normalized $SSC$ at $Q_{GM}$). However, from work in progress, we know that the rating exponent $b$ (which is the focus on this paper) is not affected by the long-term changes, but only the uncertainty in the estimate of $b$ will increase.

The use of commercial coffee filters, with a rather large and not clearly defined mean pore diameter, is certainly not optimal for the measurement of SSC. Comparisons of annual suspended sediment load estimated based on the coffee filters and cellulose-acetate filters, with a well-defined pore diameter of 0.45 μm, indicate that measurements based on coffee filters are underestimated by 20%, which is on the order of the clay fraction of the suspended sediment (Hillebrand et al., 2015). Since the clay fraction does not change as a function of discharge (Hillebrand and Frings, 2017), we assume that the rating analysis is not affected by the choice of the filters; the limitations that are associated to the larger uncertainty of single SSC-estimates are compensated by the larger number of measurements, which were feasible do to the low-cost filter system. This assumption is supported by the use of the glass fibre filters for the two LOI-stations at the Rhine and the Moselle, which show the same rating behaviour as the suspended monitoring stations (see chapter 4.2). Furthermore, the monitoring approach did not change during the monitoring period and therefore long-term changes due to the sampling and lab analysis can be discarded.

### 4.1 Controls on rating behaviour of suspended sediment

The sediment rating concept, which expresses suspended sediment concentration ($SSC$) or suspended sediment load ($Q_s$) as a function of discharge ($Q$), is based on the assumption that factors controlling the generation of runoff in the catchment are closely linked with factors controlling the sediment supply to the river channel (Gray, 2018). This is certainly the case if rainfall produces erosive surface runoff, which in turn results in sheet, rill and/or gully erosion (e.g. Poesen, 2018) and the presence of this process chain is supported by the majority of the rating curves presented in this study: 51 of 62 stations show a clear increasing trend with a positive rating exponent in the high flow regime, which is attributed to the production of surface runoff and strong sediment supply through sheet and rill erosion. A positive rating exponent of the $SSC$-$Q$ relation implies that the sediment load increases "faster" than the discharge (e.g. sediment load increases more than twofold if discharge doubles). This follows from the following equation:

$$Q_s = SSC \times Q = (aQ^b) \times Q = aQ^{b+1} \qquad (3),$$

which indicates a rating exponent > 1 for the $Q_s$-$Q$ relation for $b > 0$, as shown for most stations. However, if sediment load increases "faster" than the discharge, additional sediment sources (either external or internal) must be mobilized as discharge increases. Rivers showing rapid increases of $SSC$ (and thus $Q_s$) are termed "reactive" rivers by Syvitski et al. (2000). The (re-)activation of sediment sources can be mainly explained by the extension of areas of water-saturated soils, which

355 contribute to surface runoff and discharge and thus increases the connectivity during rainstorm events (Bracken et al., 2013; Fryirs, 2013). Since topography (especially hillslope gradient, path lengths and surface roughness) exerts a dominant control on hydrological and sediment connectivity (Baartman et al., 2013; Heckmann et al., 2018; Hoffmann, 2015), a strong relationship between the rating exponent and the topographic characteristics of the catchments can be anticipated (Gray, 2018; Syvitski et al., 2000). Our results show a clear trend of increasing $b_h$ as the fraction of steep hillslope (i.e. slope

gradients > 10%) increases, thus confirming the expectation. Thus high $b_h$ values are observed at gauging stations with discharge contributions from the European Alps (e.g. the Danube below Jochenstein and the Rhine) and from tributaries with mountainous catchment topography (e.g. the Neckar and Moselle catchments). The strong control of the slope gradient of the contributing catchments indicates that additional sediment sources, which are mobilized during increasing discharges, are primarily located on hillslopes (i.e. external sediment sources) and sediment sources within the river play a minor role. Our

results that show steeper rating curves for the Rhine tributaries than the Rhine itself are confirmed by the results by Asselmann (2000), which were obtained from a limited number of stations, with tributaries showing steeper rating curves than the larger river Rhine. In Taiwan, Hilton et al. (2012) observed a similar trend for the gradient of the relationship between particulate organic carbon (POC) concentration and discharge, which increased with the proportion of catchment area steeper than 35°. This hillslope-gradient is frequently considered as a threshold for mass wasting and erosion process,

indicating that additional sources in mountain systems are provided as threshold hillslopes become more widespread.

While $b_h$ is dominantly controlled by topography, Fig. 7b indicates also a climatic control on the rating exponent in the high flow regime. Interestingly, most stations from the Elbe and Oder catchments plot below the regression line in Fig. 7b. This indicates that the Elbe and Oder show lower $b_h$ values for a similar fraction of slopes steeper than 10% compared to the general trend. Assuming similar catchment topographies for a specified percentage of catchment area steeper 10%, the lower

$b_h$ values are mainly explained by climatic differences. The dry continental climate in the Elbe and the Oder catchments likely reduces the reactivity of the river systems, requiring larger increases of rain and discharge to increase the specific sediment supply in these basins compared to basins with higher/more frequent precipitation in the western part of Germany. The lower reactivity may be explained by the general tendency of lower antecedent soil moisture in more continental climates, and thus a slower increase of water-saturated soils, that increase the sediment connectivity during rainstorm events

(Bracken et al., 2013; Fryirs, 2013). Furthermore, sandstone in the Thuringia Forest in the Elbe headwater and extensive glacio-fluvial deposits along the Elbe and the Oder may be more porous and generate less runoff than the schists in mountains and highlands along the Rhine and in central west Germany.

The break of the rating behaviour, which is observed for 52 of the 62 suspended sediment stations along the German waterways, implies a change of processes and/or factors controlling suspended sediment in river channels at the transition

from low to high flow regimes. A similar scale break has been shown along the Rhone river in France by Poulier et al. (2019). Interestingly, the break for most stations present in this study occurs at $Q/Q_{GM} \sim 1.1$, which is roughly equivalent to $Q/Q_{avg} \sim 0.9$ or $Q/Q_{median} \sim 1.0$. Given that the break is close to the median discharge implies that river discharge is approximately 50% of the time in the low flow rating regime and 50% in the high flow rating regime.

In contrast to $b_h$, there is no simple relation of the low-flow rating exponent $b_l$ to the topographic characteristics of the contributing catchment (Fig. 7a). This result is not unexpected, given the fact that hillslopes during low flow conditions do not contribute significantly to runoff and suspended sediment in the river channel, but discharge mainly results from ground water supply. Thus, the transition from $b_l$ to $b_h$ likely reflects a change of factors controlling suspended sediment supply, which is likely associated with the transition from high authochonous organic matter content at low flow to dominantly allochthonous mineral content at high flow (compare also chapter 4.2 and 4.3).

Many of the tributary waterways of the Rhine, and the Upper Rhine itself, are controlled by barrages to support navigation during low flow and to supply energy. Thus, the operation of barrages and management of water flow is a potential factor controlling the rating break. Reservoirs upstream of the barrages act as sediment sinks for cohesive fines during low flow conditions (Hoffmann et al., 2017). During high flows, weir shutters are opened to prevent damage to the barrages and to control floods. Significant amounts of fine cohesive sediments can be potentially remobilized during high flows if critical shear stresses at the reservoir bed can exceed shear strength of the cohesive fines. However, in most cases weir shutters are only opened during floods, implying resuspension of cohesive sediments only at discharges much higher than $Q_{avg}$. Furthermore, preliminary evidence indicates that reservoirs upstream of weirs act as sediment sinks especially during high flows when large amounts of sediment are transported (Hoffmann et al., 2017). Given that the prominent rating break occurs at lower discharges (i.e. at $Q/Q_{GM} \sim 1.1$ or $Q/Q_{avg} \sim 0.9$) than those discharges which potentially resuspend cohesive sediments in the upstream reservoirs, barrage operation does not seem to control the rating break. Furthermore, the rating break is also observed in free-flowing waterways (without barrages), pointing to controlling factors not related to the management of the weirs or reservoirs.

Therefore, the question remains which factors control the rating exponent at low flows and the transition of the rating behaviour at average discharge? Our data show that the contribution of organic suspended matter to total SSC may play a crucial role of the SCC rating at low flows.

**4.2 Controls of the mineral vs. organic fraction of the suspended sediment**

Here we use *LOI* as a measure of the organic fraction of the total suspended solids. Results from the *LOI* measurements at the two stations in Koblenz show generally higher *LOI* values at the Moselle (where LOI > 0.5 is frequently observed) compared to the Rhine, where *LOI* rarely exceed 0.5 (less than 1% of all measurements). Both stations reveal a significant control of discharge on *LOI*. Negative rating exponents of $-0.51 \pm 0.03$ and $-0.47 \pm 0.01$, for the rivers Rhine and Moselle, respectively, indicate a declining organic matter fraction with increasing discharge (Fig. 8). At the Moselle, declining trends are partially explained by seasonal effects, with low discharges and high *LOI* dominating in summer and high discharges and low *LOI* dominating in winter months (Fig. 8 and Fig. 9). However, along the Rhine, seasonal trends are much less pronounced, and *LOI*-values scatter more strongly around the regression line in Fig. 8e compared to those of the

River Moselle (Fig. 8f). The decline of *LOI* with $Q$ (Fig. 8 e+f) is equivalent to a decline of *LOI* with *SSC*, as $Q$ and *SSC* are strongly related (Fig. 8a+b). These results are similar to those from global compilations of riverine *POC* (Meybeck, 1982, Ludwig and Probst 1996), where the strong decline of *POC* with *SSC* reflects the degradation of soils that increases with the mechanical erosion of the catchments (which is associated to high *SSC*). However, the rating relation of *LOI* or *POC* to *SSC* or $Q$ for single river systems is more likely linked a to shifts of OC-sources at various discharges (see for instance Gomez et

al., 2003; Goñi et al., 2014; Hilton et al., 2012; Moreira-Turcq et al., 2013). In the Erlenbach catchment (Switzerland), Smith et al. (2013) found declining POC levels with increasing discharge, which are related to the dilution of POC for $Q/Q_{mean} <$ 10. Above this threshold additional POC is supplied through the erosion of organic rich top-soils (from wetlands and alpine meadows) and $\%POC$ increases with $Q$. This shift from a negative to a positive rating curve of POC is related to a dominant supply of low-OC bedrock during low flow and a supply of OC-rich top-soils at high flow through surface runoff.

The organic carbon content of top-soils in Moselle and Rhine catchments typically ranges between 2 and 12 % (Jones et al., 2005). The values are similar to the *LOI*-values at high flows in both rivers, on the order of 0.1. Thus, negative rating exponents of *LOI* indicate that suspended matter at low flow is enriched in organic carbon and diluted when it is swamped by mineral and catchment-derived OC (with low *LOI* ~ 0.1) at high flow. Enrichment of OC at low flow highlights the primary control of low flow dynamics with increased water and plankton residence time on *LOI*. In addition to the

controlling flow dynamics, higher *LOI* during spring and summer months shows the positive effect of water temperature and light availability on plankton growth (Cloern, 1999), which may dominate the total suspended organic matter in the river Moselle at Koblenz especially during April and May (Fig. 9) (Hardenbicker et al., 2014). Under warm low flow conditions, increasing discharges rapidly dilute high concentrations of autochthonous carbon, causing a decline of total suspended sediment (which is dominated by the organic fraction under warm low-flow conditions) as evidenced in early summer 2011,

which was characterized by exceptionally low discharge of the Rhine in May and June (Hardenbicker et al., 2016). While a positive correlation between $SSC_{tot}$ and $Q$ was observed for most of the year in 2011 in the Rhine at Koblenz, $SSC_{tot}$ relates negatively with discharge during these low-flow months, indicating a shift in the SSC regime as phytoplankton dominates the organic suspended fraction.

At stations where the organic fraction of SSC generally adds a substantial share to the total SSC (e.g. as in the case of the

Moselle, where LOI reaches 60% at low flows), the rating exponent $b_l$ is negative. For instance, Hardenbicker et al. (2016) reported for the Elbe that *LOI* and *Chla* contributions to SSC increased with distance downstream that is associated with a downstream decrease of $b_l$ (Fig. 6a). Furthermore, low *LOI*-levels in the upper and middle Rhine are characterized by higher $b_l$-values (~0.5). Thus our results indicate that the suspended sediment rating at low flows is strongly controlled by intrinsic (within-channel) processes that govern the formation of organic matter within the river channel: organic rich stream flows

are generally characterized by $b_l$-values close to 0 or < 0, while organic poor channels show typically $b_l>0.5$.

### 4.3 Modelling of the total suspended sediment

The presented data indicate that the observed rating break of total suspended sediment concentration is mainly controlled by the transition from autochthonous production of organic suspended matter at low flows to the allochthones supply of (dominantly mineral) suspended matter during high flows. Our results suggest that $SSC_{mrl}$ and $LOI$ can be modelled separately using a power law rating relation. If the rating behaviour of $SSC_{mrl} = f(Q/Q_{GM})$ and $LOI = f(Q/Q_{GM})$ is known, the organic and total suspended sediment concentration can be estimated separately:

$$SSC_{org} = \frac{LOI}{1-LOI} SSC_{mrl}, \qquad SSC_{tot} = \left(\frac{LOI}{1-LOI} + 1\right) SSC_{mrl} \tag{4}$$

Using the bootstrap regression of the LOI-station at the River Moselle, with $SSC_{mrl} = (5.27 \pm 0.14) \times (Q/Q_{gm})^{(1.37\pm0.03)}$ and $LOI = (0.32 \pm 0.004) \times (Q/Q_{gm})^{(-0.47\pm0.01)}$, the modelled $SSC_{tot}$ (Fig. 10) shows the following features: i) at very low discharges ($\sim Q/Q_{gm} < 0.2$, which are rarely observed at the Moselle), $SSC_{tot}$ typically declines with increasing Q, ii) at higher discharges, $SSC_{tot}$ increases with discharge, iii) the gradient of the modelled $SSC_{tot}$-lines

continuously increases with $Q$ and approaches the rating exponent of the mineral SSC-fraction at high $Q/Q_{gm}$. This model result generally agrees with the measured $SSC_{tot}$-values. The decrease of the modelled $SSC_{org}$-values at very low discharges supports the notion that the organic fraction of the suspended matter is affected by dilution effects. The dilution effect of the autochthonous organic matter is outpaced by increased (allochthones) supply of organic matter, which leads to increasing $SSC_{org}$ at higher discharges as a result of strong supply of organic rich top soils through surface runoff and soil erosion.

Empirical sediment rating curves show distinct rating breaks slightly above $Q/Q_{gm} \sim 1$ for most stations. In contrast, suspended sediment rating relation of the modelled $SSC_{tot}$ based on Eq. 4 changes more gradually i) from negative relations at very low discharges, ii) to slight increases of $SSC_{tot}$ at low to medium (average) discharge and iii) to strong increases of $SSC_{tot}$ with $Q/Q_{gm}$ at high discharges. The gradient of the modelled $SSC_{tot}$ at high discharges approaches the rating exponent of $SSC_{mrl}$, which is similar to the rating exponent $b_h$ in the high flow domain above the rating break. Assuming

that $a_h$ and $b_h$ are mainly controlled by the mineral fraction of the suspended sediment, we argue that the rating of the high flow regime can be used as a first order approximation of the $SSC_{mrl}$ at low flow conditions and that the excess of $SSC_{tot}$ compared to the modelled $SSC_{mrl}$ is primarily explained by the organic fraction of the suspended sediment (compare Fig. 10 and 11). Differences between $b_h$ (i.e. the rating at high discharge) and the rating of $SSC_{mrl}$ may be partially explained by the organic fraction of suspended sediment that is not derived from *in situ* (autochthonous) organic matter, but is supplied from

hillslope through the erosion of organic rich top soils.

In case of the river Moselle, our results indicate that $SSC_{tot}$ exceeds $SSC_{mrl}$ by a factor of ~1.5 to 2 at discharges smaller than $Q_{GM}$. Thus, monitored suspended sediment yields, which are mostly based on estimates of the total $SSC$, overestimate the mineral fraction of the $SSC$ at low to moderate flows. The frequency analysis of the long-term suspended monitoring data at the Rhine station at Koblenz, which integrates the organic and mineral fraction of the suspended matter, shows roughly 50

% of the total annual suspended load is transported in 10% of the time during floods. Due to the inclusion of the organic

matter and the resulting overestimate of the (mineral) suspended sediment at low to medium flows, floods are likely to be more important in the transport of the mineral fraction of the suspended load than otherwise estimated. In the case of a clear rating break, our conceptual model separating the rating at low and high flows due to the shift of the process regime can be used to separate the organic and mineral fraction and give a first order estimate of the autochthonous organic fraction of the total SSC.

In the case of substantial contribution of organic SSC to the total SSC, our results suggest that the common practice of using a continuous sediment rating relation results in large errors that can be reduced. These errors potentially influence regression results at high SSCs, generally leading to an underestimate of SSC based on continuous rating curves. Much better results for the prediction of SSC and hence sediment load can be achieved by applying rating relationships that include rating breaks.

The findings are likely to be representative for other large river systems, with similar suspended sediment and nutrient loadings. However, more work is needed to see if the conceptual rating model can be applied to other large river systems with strong human interference on sediment and nutrient supply.

**5. Conclusion**

Using more than 750 000 suspended sediment and discharge measurements at 62 gauging stations along 19 waterways in Germany and more than 2000 measurements of the loss on ignition of suspended matter at two stations along the rivers Moselle and Rhine, we performed a detailed rating analysis of suspended matter and its organic content. Our main findings may be summarized as follow:

1. For most studied gauging stations, rating coefficients are not constant over the full discharge range, but show a distinct break in the sediment rating curve, with specific $SSC - Q$ domains above and below this break. Typically, the rating break occurs slightly above the geometric mean discharge.

2. The transition of the rating exponent (from $b_l$ to $b_h$) is likely a result of a change in controlling factors of suspended sediment from intrinsic (within the river system) to extrinsic (outside the river channel but within the catchment) sources. Our results suggest that in large, low-turbidity rivers the formation of organic matter within the river channel is an important control of the rating behaviour at low discharges, while the extrinsic control is related to the supply of suspended sediment due to top-soil erosion in the catchment. This hypothesis is supported by the relationship between the rating exponent and the fraction of hillslopes steeper than 10% within the contributing catchment area and LOI values of roughly 0.1 at high flow, which resemble typical top-soil SOC concentrations ranging between 2-12 % in the Moselle and Rhine catchment.

3. Based on these findings we developed a conceptual rating model for large (>10000 km$^2$) and low-turbid (SSC < 1000 mg l$^{-1}$) rivers separating the mineral and organic fraction of the suspended matter in the Germany waterways. The model assumes a positive power law rating of the mineral fraction of the SSC with Q and a negative power law rating of the LOI with Q and can be used to model the rating behaviour of the total SSC as frequently measured by

suspended monitoring networks. More work is needed to see if the conceptual rating model can be applied to other large river systems that are controlled by strong human induced sediment and nutrient supply.

*Author contributions.* HF provided OC data from both stations in Koblenz and provided feedback on data anlysis and
discussion of results. YB provided SSC data and analysed the spatial distribution of the rating coefficients. JB contributed to the rating analysis and supported discussion on the results. TH performed rating analysis of all stations and prepared the manuscript with the cooperation of all co-authors.

*Competing interests.* The authors declare that they have no conflict of interest.

*Acknowledgements.* The data used in this paper were provided by the suspended sediment monitoring network of the German waterways that was established in the 1960s by the Federal Waterways and Shipping Administration (Wasserstraßen- und Schifffahrtsverwaltung des Bundes, WSV). We acknowledge the WSV for maintaining the monitoring network and for suspended sediment sampling. Furthermore, we thank Kristin Bunte, one anonymous reviewer and the Associate Editor
Robert Hilton for their helpful comments and suggestions that greatly improved the quality of this paper.

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

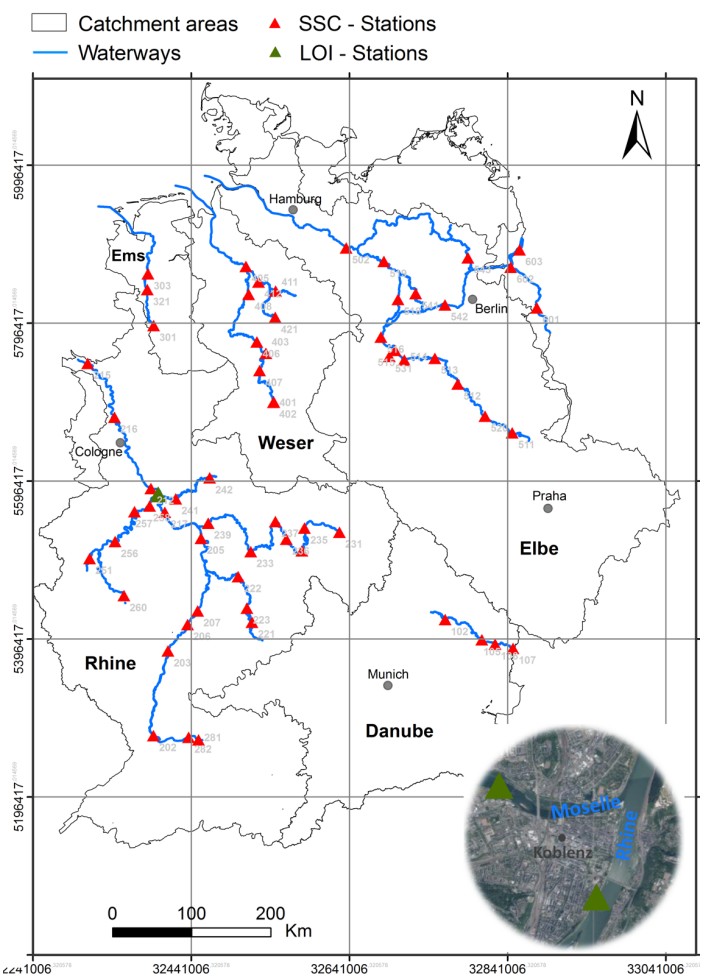

**Figure 1: Selected sampling locations of the WSV-suspended monitoring network used in this study covering the major river basins in Germany as shown. Labels refer to the station codes given in Tab. 1. The inset shows an orthophoto of the city of Koblenz, including the two LOI stations along the river Rhine and Moselle. Coordinates are given in UTM ETRS 32 8st.**

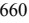

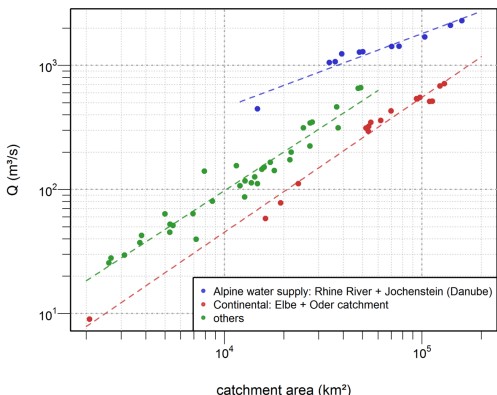

**Figure 2: Discharge as a function of catchment area for 62 gauging stations that are used as reference stations of the suspended sediment monitoring network.**

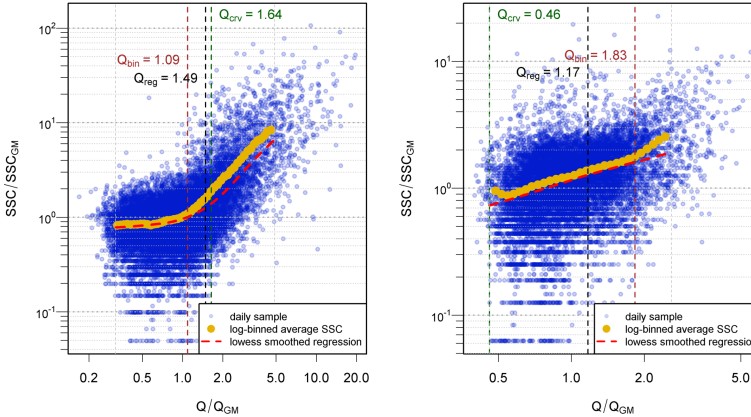

**Figure 3: Rating curves of the stations at Rockenau (river Neckar, ID 222) and Kachlet (river Danube, ID 103). For locations see Fig. 1. $Q_{crv}$, $Q_{reg}$ and $Q_{bin}$ refer to the rating breaks derived using the maximum curvature of the lowess curve, the regression of the data points of the loess curve and the regression of the binned averages (formore detailed see text).**

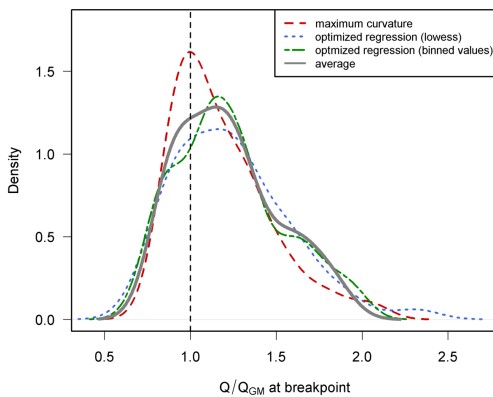


**Figure 4: Density distribution of rating breaks ($Q_{br}$) derived from the scaling analysis of the suspended sediment concentration. For detailed results see also Tab. 2.**

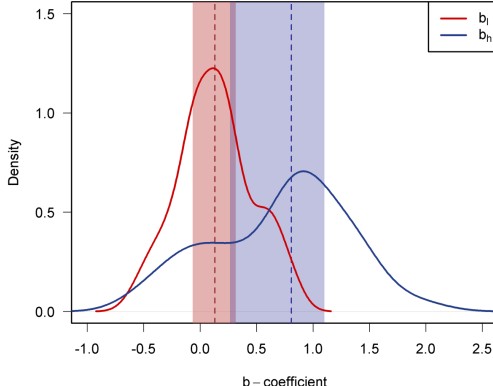

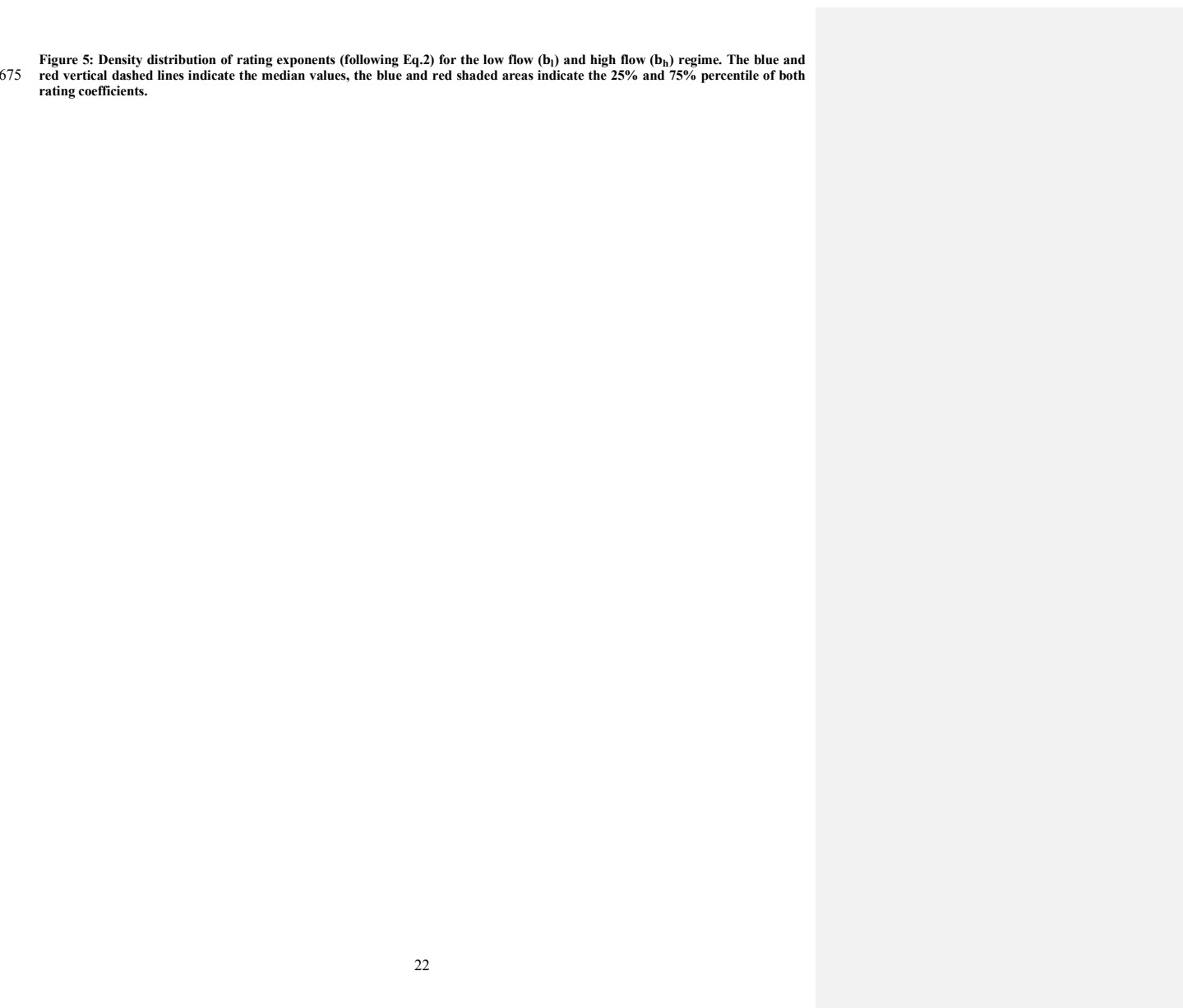

**Figure 5: Density distribution of rating exponents (following Eq.2) for the low flow ($b_l$) and high flow ($b_h$) regime. The blue and red vertical dashed lines indicate the median values, the blue and red shaded areas indicate the 25% and 75% percentile of both rating coefficients.**


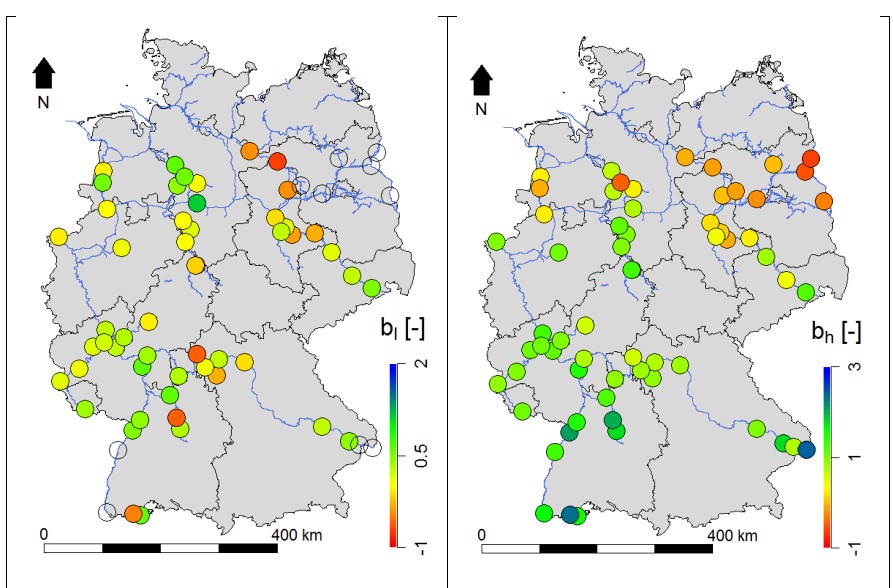

**Figure 6: Maps representing the spatial distribution of the rating exponents. Left map indicates $b_l$ and right map indicates $b_h$. Empty circles in the left graph denote stations without a significant rating break.**

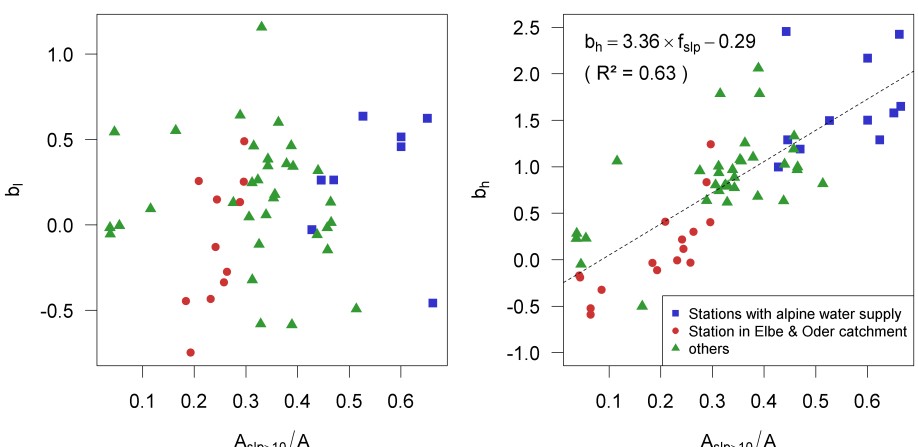

**Figure 7: Relation of rating coefficients ($b_l$ left and $b_h$ right) and the ratio of the catchment area with hillslopes steeper than 10% ($A_{slp>10}$) vs. total catchment area ($A$).**

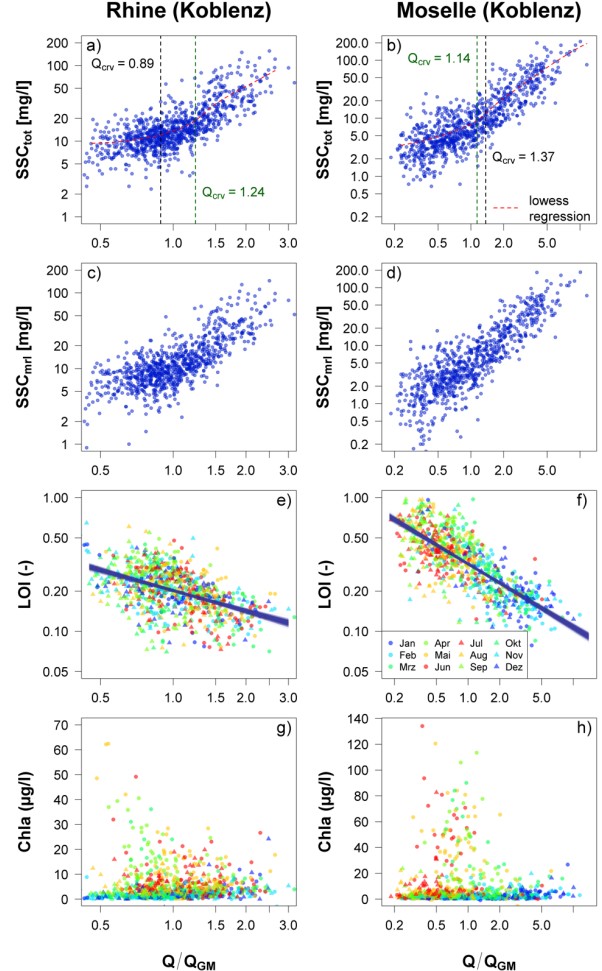


**Figure 8: Rating of total SSC (1st row), mineral SSC (2nd row), loss on ignition (LOI, 3rd row) and chlorophyll a (Chla, 4th row) for the station Koblenz-Rhine (left) and Koblenz-Moselle (right). Blue lines on LOI-scatter plots show regression results using 1000 bootstrap-replicates.**

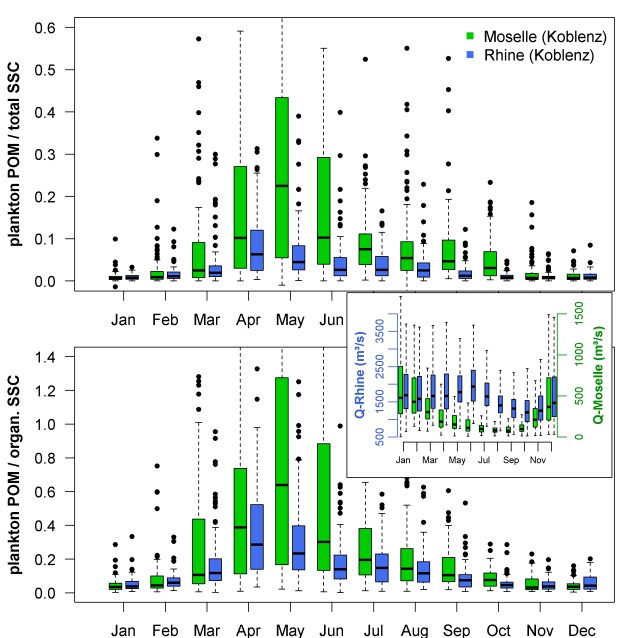

**Figure 9: Boxplot of seasonal variation of plankton POM with respect to total *SSC* (top) and organic *SSC* (bottom) for the stations Koblenz Rhine (blue) and Koblenz Moselle (green) from 1990-2017. Organic *SSC* is derived from *LOI* measurements and plankton *POM* is calculated from *Chla*-measurements using a *POC/Chla*-ratio of 40 and a *POC/POM*-ratio of 0.42. Plankton POM/organic SSC ratios > 1 are due to measurement errors of *Chla* and *LOI* and due to simplified conversion ratios. The inset shows the boxplot of seasonal discharge variations at both stations.**


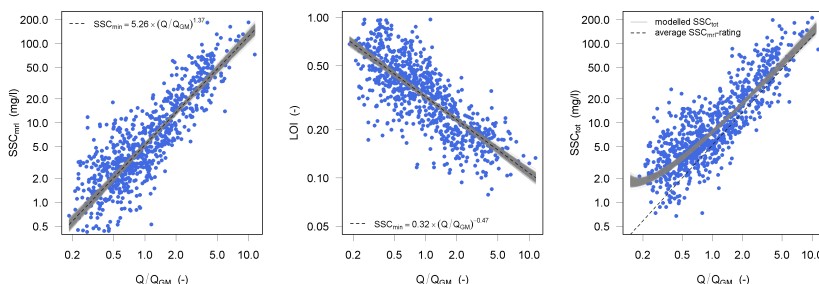

**Figure 10: Statistical modelling of the total suspended sediment concentration ($SSC_{tot}$, right) based on the positive and negative power law rating of the mineral fraction of the SSC ($SSC_{mrl}$, left) and the loss on ignition ($LOI$, middle), respectively.**


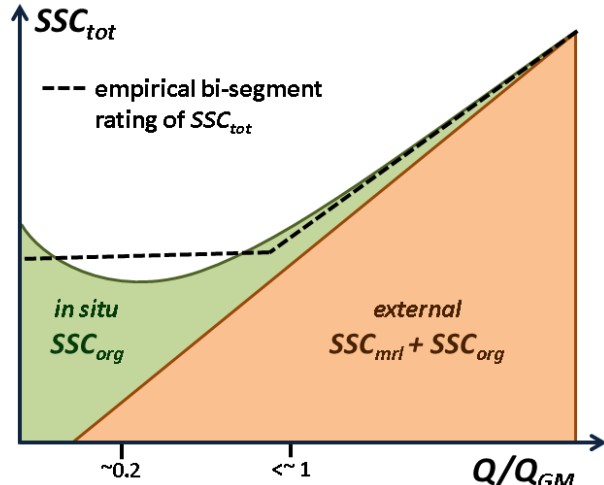

**Figure 11: Conceptual model of suspended sediment rating in the German waterways.**


**Table 1: Overview of sampling locations, contributing catchment size, monitoring period, number of samples (n) and average (avg), median (med) and geometric mean (GM) of discharge Q and suspended sediment concentration (SSC). The map index refers to the numbers in the map (Figure 1). The Location (river-km) refers to the official kilometrage of the German Water and Shipping Authority of the water ways in downstream distance. The monitoring period refers to the start year, if only one year is given. In this case monitoring continues until present.**


| Map index | Station name | River | Location (river-km) | Catchment (km²) | Monitoring period | n | Q (mg/l) | | | SSC (mg/l) | | |
|---|---|---|---|---|---|---|---|---|---|---|---|---|
| | | | | | | | avg | med | GM | avg | med | GM |
| 102 | Straubing | Donau | 2321.3 | 37026 | 1982 | 7197 | 462 | 395 | 414 | 16.9 | 14 | 13.5 |
| 105 | Vilshofen | Donau | 2249.5 | 47609 | 1966 | 11282 | 651 | 573 | 597 | 20.0 | 16 | 15.3 |
| 106 | Kachlet | Donau | 2230.7 | 49045 | 1975 | 9214 | 661 | 578 | 606 | 20.4 | 17 | 16.1 |
| 107 | Jochenstein | Donau | 2203.1 | 76653 | 1974 | 9174 | 1425 | 1292 | 1319 | 47.9 | 23 | 25.2 |
| 281 | Reckingen | Rhein | 90.2 | 14718 | 1972 | 10038 | 446 | 411 | 416 | 13.9 | 10 | 10.6 |
| 282 | Albbruck Dogern | Rhein | 108.9 | 33987 | 1972 | 10601 | 1053 | 969 | 974 | 19.7 | 13 | 13.0 |
| 202 | Weil | Rhein | 173 | 36472 | 1970 | 10104 | 1070 | 984 | 985 | 27.5 | 17 | 17.0 |
| 203 | Kehl | Rhein | 294 | 39330 | 1970-2013 | 11201 | 1237 | 1134 | 1145 | 16.4 | 11 | 11.4 |
| 206 | Plittersdorf | Rhein | 339.8 | 48276 | 1977-2013 | 9281 | 1278 | 1170 | 1187 | 18.3 | 14 | 14.2 |
| 207 | Maxau | Rhein | 362.3 | 50196 | 1964 | 13203 | 1285 | 1175 | 1189 | 22.9 | 18 | 16.8 |
| 205 | Nierstein | Rhein | 480.6 | 70387 | 1983 | 7228 | 1419 | 1283 | 1315 | 21.1 | 17 | 17.1 |
| 217 | Sankt Goar | Rhein | 557 | 103488 | 1970 | 11398 | 1694 | 1520 | 1557 | 26.3 | 21 | 20.3 |
| 215 | Emmerich | Rhein | 851.9 | 159555 | 1982 | 8710 | 2289 | 1950 | 2072 | 27.2 | 25 | 22.7 |
| 212 | Weißenthurm | Rhein | 608.2 | 139549 | 1971 | 9919 | 2102 | 1830 | 1893 | 32.9 | 25 | 25.9 |
| 221 | Poppenweiler | Neckar | 164.9 | 5005 | 1965-2014 | 11609 | 64 | 45 | 49 | 30.7 | 18 | 17.5 |
| 222 | Rockenau | Neckar | 61.3 | 7916 | 1971 | 10364 | 140 | 104 | 109 | 34.7 | 19 | 20.3 |
| 223 | Lauffen | Neckar | 125.1 | 12676 | 1987 | 6209 | 87 | 65 | 69 | 22.0 | 15 | 14.8 |
| 231 | Viereth | Main | 380.8 | 12010 | 1972-2005 | 8455 | 107 | 78 | 86 | 18.6 | 16 | 15.2 |
| 235 | Garstadt | Main | 323.7 | 12722 | 1986-2005 | 4949 | 117 | 82 | 93 | 23.4 | 19 | 18.7 |
| 232 | Marktbreit | Main | 275.7 | 13693 | 1965-2012 | 11801 | 113 | 83 | 91 | 25.8 | 21 | 19.5 |
| 236 | Erlabrunn | Main | 241.2 | 14244 | 1986-2005 | 4941 | 126 | 90 | 101 | 25.3 | 21 | 20.9 |
| 237 | Steinbach | Main | 210 | 17914 | 1987 | 6121 | 142 | 103 | 114 | 20.3 | 17 | 16.2 |
| 233 | Kleinheubach | Main | 121.7 | 21505 | 1973-2014 | 9702 | 173 | 124 | 136 | 27.6 | 22 | 21.0 |
| 239 | Eddersheim | Main | 15.6 | 27100 | 1986-2012 | 6506 | 223 | 161 | 177 | 28.9 | 23 | 22.8 |
| 242 | Wetzlar | Lahn | 125.3 | 2669 | 1986-2007 | 4843 | 28 | 16 | 18 | 20.9 | 17 | 15.9 |
| 241 | Kalkofen | Lahn | 31.6 | 5303 | 1970 | 10240 | 45 | 26 | 30 | 22.2 | 14 | 13.4 |
| 277 | Hamm. Wehr | Lippe | 120.1 | 2607 | 1976 | 8737 | 26 | 19 | 20 | 15.6 | 12 | 11.7 |
| 260 | Güdingen | Saar | 91.7 | 3811 | 1973 | 10314 | 42 | 26 | 29 | 18.3 | 11 | 10.7 |
| 251 | Wincheringen | Mosel | 221.9 | 11522 | 1974 | 9558 | 155 | 91 | 99 | 31.7 | 22 | 23.2 |
| 256 | Detzem | Mosel | 166.8 | 25130 | 1981-2002 | 5320 | 314 | 183 | 205 | 31.7 | 21 | 22.3 |
| 257 | Cochem | Mosel | 50.2 | 27165 | 1981-2011 | 7387 | 343 | 211 | 231 | 29.4 | 19 | 21.5 |
| 258 | Brodenbach | Mosel | 27.2 | 27872 | 1981-2009 | 6672 | 349 | 214 | 235 | 31.9 | 21 | 22.4 |

| 321 | Meppen | Hase | 1 | 3126 | 1974-1996 | 5740 | 29 | 22 | 24 | 21.9 | 21 | 19.4 |
|-----|--------|------|---|------|-----------|------|----|----|----|------|----|------|
| 301 | Rheine | Ems | 153 | 3740 | 1964 | 13506 | 37 | 23 | 25 | 27.4 | 18 | 17.4 |
| 303 | Lathen | Ems | 253.3 | 8696 | 1966 | 12951 | 80 | 57 | 61 | 18.6 | 16 | 14.6 |
| 421 | Herrenhausen | Leine | 87.1 | 5304 | 1965-2006 | 10448 | 52 | 38 | 41 | 40.0 | 24 | 23.8 |
| 411 | Marklendorf | Aller | 75.9 | 7209 | 1971 | 11563 | 40 | 30 | 32 | 14.9 | 14 | 12.5 |
| 412 | Rethem | Aller | 34.2 | 14730 | 1973 | 10601 | 111 | 85 | 91 | 21.4 | 19 | 17.6 |
| 401 | Hann.-Münden.W. | Werra | 0.5 | 5497 | 1965 | 12639 | 51 | 38 | 40 | 51.6 | 39 | 39.0 |
| 402 | Hann.-Münden.F. | Fulda | 1 | 6947 | 1965 | 12639 | 64 | 43 | 51 | 23.3 | 18 | 16.0 |
| 407 | Höxter | Weser | 69.4 | 15501 | 1983 | 8224 | 145 | 103 | 118 | 30.4 | 23 | 23.0 |
| 403 | Bodenwerder | Weser | 110.7 | 15924 | 1964 | 12875 | 151 | 109 | 123 | 32.3 | 24 | 23.7 |
| 406 | Hameln | Weser | 135.2 | 17077 | 1979 | 9333 | 166 | 118 | 134 | 32.0 | 24 | 23.6 |
| 408 | Nienburg | Weser | 268.1 | 21815 | 1985 | 7862 | 199 | 143 | 163 | 29.5 | 23 | 22.7 |
| 405 | Intschede | Weser | 329.5 | 37720 | 1969 | 11270 | 313 | 234 | 256 | 35.1 | 28 | 27.3 |
| 543 | Zehdenick | Havel | 15.1 | 2076 | 1991 | 4290 | 9 | 7.5 | 7.1 | 10.7 | 9 | 8.7 |
| 542 | Ketzin | Havel | 34.1 | 16173 | 1991-2016 | 6310 | 58 | 54 | 45 | 12.9 | 11 | 9.8 |
| 541 | Rathenow | Havel | 103.6 | 19288 | 1991-2016 | 6498 | 78 | 72 | 64 | 14.9 | 13 | 11.8 |
| 531 | Calbe | Saale | 20 | 23719 | 1991 | 6740 | 111 | 84 | 93 | 26.8 | 20 | 21.7 |
| 511 | Pirna | Elbe | 34.7 | 52080 | 1991 | 6120 | 313 | 234 | 257 | 22.7 | 18 | 16.8 |
| 520 | Meissen | Elbe | 83.4 | 53885 | 1994 | 5310 | 323 | 239 | 264 | 25.1 | 19 | 19.8 |
| 512 | Torgau | Elbe | 154 | 55211 | 1993 | 5790 | 346 | 253 | 282 | 32.1 | 27 | 26.1 |
| 513 | Wittenberg | Elbe | 216.3 | 61879 | 1991 | 6339 | 360 | 272 | 295 | 28.0 | 24 | 23.8 |
| 514 | Aken | Elbe | 274.8 | 69849 | 1991 | 5887 | 429 | 326 | 353 | 25.3 | 23 | 21.8 |
| 515 | Barby | Elbe | 294.8 | 94060 | 1991 | 6431 | 538 | 401 | 447 | 33.1 | 28 | 27.9 |
| 516 | Magdeburg Strombr. | Elbe | 326.6 | 94942 | 1992 | 6395 | 537 | 397 | 447 | 26.0 | 21 | 20.8 |
| 518 | Tangermünde | Elbe | 389.1 | 97780 | 1991 | 6397 | 552 | 422 | 462 | 31.2 | 27 | 26.3 |
| 519 | Wittenberge | Elbe | 454.6 | 123532 | 1993 | 5907 | 681 | 526 | 576 | 32.0 | 25 | 26.1 |
| 502 | Hitzacker | Elbe | 522.6 | 129877 | 1963 | 13703 | 712 | 571 | 605 | 34.0 | 30 | 28.4 |
| 601 | Frankfurt / Oder | Oder | 585.8 | 53590 | 1991 | 4822 | 294 | 246 | 252 | 24.8 | 21 | 20.6 |
| 602 | Hohensaaten | Oder | 662.3 | 109564 | 1991 | 4943 | 512 | 440 | 453 | 21.7 | 18 | 17.6 |
| 603 | Schwedt | Oder | 690.6 | 112950 | 1991 | 5022 | 513 | 442 | 454 | 23.8 | 20 | 19.4 |

**Table 2: Results from rating-break analysis and and log-linear and non-linear least square regression of rating exponent (Eq. 2) above ($b_h$) and below ($b_l$) rating break ($Q_{br}/Q_{gm}$). $\Delta b_l$ and $\Delta b_h$ refer to the uncertainty of the rating breaks derived from the bootstrap analysis.**

| Map index | name | $Q_{br}/Q_{gm}$ | | | | log-linear regression | | | | non-linear LS regression | | | |
|---|---|---|---|---|---|---|---|---|---|---|---|---|---|
| | | loess-regression | loess-curvature | binned regression | mean | $b_l$ | $\Delta b_l$ | $b_h$ | $\Delta b_h$ | $b_l$ | $\Delta b_l$ | $b_h$ | $\Delta b_h$ |
| 102 | Straubing | 1.03 | 1.09 | 1.09 | 1.07 | 0.28 | 0.04 | 0.97 | 0.04 | 0.25 | 0.08 | 1.01 | 0.08 |
| 105 | Vilshofen | 1.55 | 1.55 | 1.54 | 1.55 | 0.65 | 0.03 | 1.50 | 0.07 | 0.46 | 0.07 | 1.79 | 0.11 |
| 106 | Kachlet | - | - | - | - | - | - | 0.68 | 0.02 | - | - | 0.74 | 0.03 |
| 107 | Jochenstein | - | - | - | - | - | - | 2.01 | 0.02 | - | - | 2.45 | 0.15 |
| 202 | Weil | - | - | - | - | - | - | 1.23 | 0.02 | - | - | 1.65 | 0.22 |
| 203 | Kehl | - | - | - | - | - | - | 0.85 | 0.02 | - | - | 1.29 | 0.08 |
| 205 | Nierstein | 1.09 | 1.05 | 1.15 | 1.1 | 0.65 | 0.04 | 1.44 | 0.04 | 0.64 | 0.07 | 1.50 | 0.09 |
| 206 | Plittersdorf | 1.31 | 1.29 | 1.37 | 1.32 | 0.69 | 0.04 | 1.80 | 0.07 | 0.52 | 0.09 | 2.17 | 0.15 |
| 207 | Maxau | 0.95 | 0.97 | 0.87 | 0.93 | 0.60 | 0.04 | 1.41 | 0.03 | 0.46 | 0.06 | 1.50 | 0.05 |
| 212 | Weißenthurm | 0.95 | 0.99 | 0.87 | 0.94 | 0.36 | 0.03 | 1.18 | 0.03 | 0.26 | 0.07 | 1.29 | 0.05 |
| 215 | Emmerich | 1.08 | 1.11 | 1.15 | 1.11 | -0.02 | 0.04 | 0.93 | 0.04 | -0.03 | 0.05 | 1.00 | 0.05 |
| 217 | Sankt Goar | 1.01 | 1.01 | 1.03 | 1.02 | 0.21 | 0.04 | 1.22 | 0.03 | 0.26 | 0.06 | 1.19 | 0.06 |
| 221 | Poppenweiler | 1.61 | 1.72 | 1.83 | 1.72 | 0.23 | 0.02 | 1.52 | 0.05 | 0.34 | 0.12 | 1.79 | 0.08 |
| 222 | Rocke-u | - | - | 1.09 | 1.09 | 0.15 | 0.02 | 1.37 | 0.03 | 0.60 | 0.09 | 1.26 | 0.05 |
| 223 | Lauffen | 1.33 | 1.39 | 1.63 | 1.45 | 0.12 | 0.03 | 1.31 | 0.06 | -0.59 | 0.19 | 2.06 | 0.13 |
| 231 | Viereth | 1.28 | 1.15 | 1.09 | 1.17 | -0.01 | 0.03 | 0.70 | 0.03 | -0.11 | 0.10 | 0.80 | 0.09 |
| 232 | Marktbreit | 1.01 | 1.02 | 1.09 | 1.04 | -0.26 | 0.03 | 0.87 | 0.03 | -0.32 | 0.05 | 0.94 | 0.05 |
| 233 | Kleinheubach | 1.17 | 1.12 | 1.3 | 1.2 | 0.25 | 0.03 | 0.92 | 0.03 | 0.35 | 0.04 | 0.89 | 0.04 |
| 235 | Garstadt | 1.27 | - | 1.3 | 1.29 | 0.25 | 0.04 | 0.74 | 0.04 | 0.26 | 0.06 | 0.79 | 0.06 |
| 236 | Erlabrunn | 1.19 | - | 1.3 | 1.25 | 0.13 | 0.03 | 0.70 | 0.04 | 0.05 | 0.06 | 0.80 | 0.06 |
| 237 | Steinbach | 0.79 | 0.84 | 0.82 | 0.82 | -0.43 | 0.05 | 0.50 | 0.03 | -0.58 | 0.08 | 0.62 | 0.06 |
| 239 | Eddersheim | 1.44 | 1.41 | 1.15 | 1.33 | 0.34 | 0.03 | 0.77 | 0.04 | 0.39 | 0.05 | 0.77 | 0.05 |
| 241 | Kalkofen | 1.94 | 1.83 | 1.54 | 1.77 | -0.11 | 0.03 | 1.34 | 0.04 | 0.32 | 0.05 | 1.03 | 0.05 |
| 242 | Wetzlar | 1.64 | 1.19 | 1.37 | 1.4 | -0.25 | 0.03 | 0.72 | 0.03 | -0.06 | 0.04 | 0.63 | 0.04 |
| 251 | Wincheringen | 1.35 | - | 1.3 | 1.33 | 0.01 | 0.01 | 1.05 | 0.02 | 0.13 | 0.03 | 0.96 | 0.02 |
| 256 | Detzem | 1.19 | - | 1.03 | 1.11 | -0.08 | 0.02 | 0.98 | 0.03 | 0.06 | 0.05 | 0.97 | 0.04 |
| 257 | Cochem | 1.19 | 1.31 | 1.22 | 1.24 | 0.18 | 0.02 | 1.01 | 0.02 | 0.16 | 0.05 | 1.08 | 0.04 |
| 258 | Brodenbach | 1.27 | 1.37 | 1.22 | 1.29 | 0.17 | 0.02 | 1.00 | 0.02 | 0.18 | 0.05 | 1.06 | 0.05 |
| 260 | Güdingen | 2.31 | 2.05 | 1.15 | 1.84 | 0.29 | 0.03 | 0.74 | 0.06 | 0.36 | 0.24 | 1.10 | 0.21 |
| 277 | Hamm. Wehr | 1.33 | 1.32 | 1.37 | 1.34 | 0.07 | 0.02 | 1.07 | 0.03 | 0.09 | 0.04 | 1.06 | 0.05 |
| 281 | Reckingen | 1.15 | - | - | 1.15 | 0.81 | 0.04 | 0.96 | 0.06 | 0.62 | 0.12 | 1.58 | 0.18 |
| 282 | Albbruck Dogern | - | - | 0.87 | 0.87 | 0.80 | 0.05 | 1.40 | 0.04 | -0.46 | 0.39 | 2.42 | 0.27 |

| | | | | | | | | | | | | | |
|---|---|---|---|---|---|---|---|---|---|---|---|---|---|
| 301 | Rheine | 1.3 | 1.05 | 1.15 | 1.17 | -0.05 | 0.02 | 0.26 | 0.03 | -0.02 | 0.04 | 0.23 | 0.03 |
| 303 | Lathen | 0.8 | 0.91 | 0.87 | 0.86 | -0.11 | 0.03 | 0.27 | 0.02 | -0.05 | 0.02 | 0.28 | 0.02 |
| 321 | Meppen | 1.14 | - | 1.15 | 1.15 | 0.61 | 0.02 | -0.11 | 0.03 | 0.54 | 0.03 | -0.05 | 0.03 |
| 401 | Hann.-Münden.W. | 1.18 | 1.18 | 1.09 | 1.15 | -0.46 | 0.02 | 0.77 | 0.03 | -0.49 | 0.06 | 0.82 | 0.06 |
| 402 | Hann.-Münden.F. | 1.54 | 1.29 | 1.94 | 1.59 | -0.06 | 0.03 | 1.11 | 0.05 | -0.15 | 0.09 | 1.33 | 0.11 |
| 403 | Bodenwerder | 1.61 | - | 1.63 | 1.62 | 0.08 | 0.03 | 0.83 | 0.05 | 0.13 | 0.06 | 0.97 | 0.08 |
| 405 | Intschede | - | - | 1.22 | 1.22 | 0.49 | 0.02 | 0.70 | 0.03 | 0.64 | 0.03 | 0.64 | 0.03 |
| 406 | Hameln | 1.84 | 1.56 | 1.73 | 1.71 | -0.07 | 0.03 | 1.17 | 0.05 | -0.02 | 0.06 | 1.19 | 0.08 |
| 407 | Höxter | 1.33 | 1.28 | 1.3 | 1.3 | -0.06 | 0.03 | 0.95 | 0.04 | 0.01 | 0.07 | 1.00 | 0.07 |
| 408 | Nienburg | 1.62 | 1.52 | 1.73 | 1.62 | 0.31 | 0.03 | 0.79 | 0.05 | 0.46 | 0.04 | 0.68 | 0.05 |
| 411 | Marklendorf | 0.76 | 0.99 | 0.73 | 0.83 | 0.00 | 0.03 | 0.31 | 0.02 | 0.00 | 0.03 | 0.23 | 0.01 |
| 412 | Rethem | 1.51 | - | 1.63 | 1.57 | 0.52 | 0.02 | -0.44 | 0.03 | 0.55 | 0.03 | -0.50 | 0.04 |
| 421 | Herrenhausen | 0.92 | 1.21 | 0.77 | 0.97 | 0.54 | 0.03 | 0.88 | 0.03 | 1.16 | 0.06 | 0.79 | 0.03 |
| 502 | Hitzacker | 1.26 | 1.08 | 1.3 | 1.21 | -0.41 | 0.02 | -0.06 | 0.03 | -0.45 | 0.02 | -0.03 | 0.03 |
| 511 | Pirna | 1.8 | - | 1.94 | 1.87 | 0.40 | 0.03 | 1.39 | 0.07 | 0.49 | 0.06 | 1.24 | 0.11 |
| 512 | Torgau | 1.46 | - | 1.45 | 1.46 | 0.15 | 0.03 | 0.86 | 0.04 | 0.13 | 0.04 | 0.83 | 0.05 |
| 513 | Wittenberg | 0.87 | 0.91 | 0.87 | 0.88 | -0.16 | 0.04 | 0.26 | 0.02 | -0.28 | 0.04 | 0.30 | 0.02 |
| 514 | Aken | 0.92 | 0.93 | 0.82 | 0.89 | -0.25 | 0.04 | -0.04 | 0.03 | -0.34 | 0.04 | -0.03 | 0.03 |
| 515 | Barby | 0.92 | 0.88 | 0.87 | 0.89 | 0.13 | 0.04 | 0.08 | 0.03 | 0.15 | 0.05 | 0.12 | 0.02 |
| 516 | Magdeburg Strombr. | 0.97 | 0.97 | - | 0.97 | -0.04 | 0.05 | 0.19 | 0.03 | -0.13 | 0.04 | 0.22 | 0.03 |
| 518 | Tangermünde | 0.93 | 0.91 | 0.92 | 0.92 | -0.28 | 0.04 | -0.06 | 0.02 | -0.43 | 0.04 | -0.01 | 0.03 |
| 519 | Wittenberge | 1.04 | 0.94 | 1.03 | 1 | -0.57 | 0.04 | -0.14 | 0.03 | -0.75 | 0.04 | -0.11 | 0.03 |
| 520 | Meissen | 0.78 | 0.86 | - | 0.82 | 0.21 | 0.05 | 0.36 | 0.03 | 0.25 | 0.06 | 0.40 | 0.04 |
| 531 | Calbe | 0.87 | 0.92 | - | 0.9 | 0.11 | 0.04 | 0.32 | 0.03 | 0.26 | 0.06 | 0.41 | 0.04 |
| 541 | Rathenow | - | - | - | - | - | - | -0.22 | 0.01 | - | - | -0.17 | 0.01 |
| 542 | Ketzin | - | - | - | - | - | - | -0.22 | 0.01 | - | - | -0.19 | 0.01 |
| 543 | Zehdenick | - | - | - | - | - | - | 0.04 | 0.01 | - | - | 0.04 | 0.02 |
| 601 | Frankfurt / Oder | - | - | - | - | - | - | -0.24 | 0.02 | - | - | -0.33 | 0.02 |
| 602 | Hohensaaten | - | - | - | - | - | - | -0.49 | 0.02 | - | - | -0.52 | 0.02 |
| 603 | Schwedt | - | - | - | - | - | - | -0.53 | 0.02 | - | - | -0.59 | 0.02 |
| 999 | Koblenz (Rhein) | 0.96 | 0.97 | | 0.96 | | | | | 0.29 | 0.20 | 2.26 | 0.18 |
| 998 | Koblenz (Mosel) | 0.87 | 0.94 | | 0.91 | | | | | -0.03 | 0.07 | 1.54 | 0.14 |
