# Peer review of "Scale-breaks of suspended sediment rating in large rivers in Germany induced by organic matter"

_Earth Surface Dynamics, 2020_

## Referee Comment (RC1) · Kristin Bunte (Referee) · 13 Mar 2020

The manuscript presents an interesting study and is well worth publishing in e-surf. Teasing apart the different sedimentary constituents (mineral, organic) that determine rating relation between suspended sediment and discharge is an important contribution. The authors quantify the different components and their change with discharge and provide a model that appears to be applicable to streams in Germany.

The description of the three methods used to compute the breakpoint flow is a bit unclear, a bit more help is needed to enable the reader the follow all steps. More of a problem is that the authors repeatedly refer to a bootstrapping approach, but never explain why bootstrapping is undertaken, what data are used, and what the purpose of the bootstrapping approach is in the first place. This needs to be revised. The manuscript also needs clarifications all over. Most of the issues are minor and can be easily addressed. I have provided a lot of suggestions for the authors when tackling those issues.

The title reflects the topic of the study, and the abstract summarizes the study effectively. The manuscript is generally well structured. An exception is the discussion Section 4.1 which is weaker than the other chapters. Items in Section 4.1 are discussed one after the other without connection, without introduction, and without a stated course of argument in the beginning.

Use of the English language is fair. The manuscript needs editorial improvement, the majority of which are minor corrections that can be easily addressed.

Several of the figures need improvement; Again, nothing serious, but revision would help to improve the manuscript quality.

In all, my evaluation of the manuscript is: publish with minor revisions, of which there are a lot, but most of them can be addressed in a straightforward way.

**Comments requesting clarification:**

L. 15: … identify discharge depended process regimes of suspended matter…. Please define better. Too short - jargon.

L 21: …into the first order control of discharge dynamics of suspended sediments. Sounds like jargon. Please start more clearly.

L 63: Please define more clearly which Q is meant. Instant Q? Mean daily Q?

L 77: The authors get too involved in describing their plot. I think I know what the authors mean, but I think they could describe this better. SR: A first look at the plot of measured values of SSC vs Q in which data are not segregates by time or processes controlling SSC exhibits strong scatter.

L 80: The authors might add that changing water supply or flow hydraulics could also be at play.

L 86: The authors should add as a 4th item to the list the effects of organic material on SSC that were introduced earlier. Doing so would also give the authors the opportunity to create a connection to the thought starting in line 86 which, as written, appears out of context. SR: Recently, we have learned more about the effects of organic material on SSC, but so far, "Most monitoring studies…

L101: Please clarify: Do the "daily" discharge measurements reflect once-a-day instantaneous measurements or are daily averages computed from or continuous measurements of Q and SSC?

L 116: Here, we selected….Please explain the reasoning for the selection

L 123: …SSC was given… do you mean "computed"

L 128: (e.g., medium and finer clays?)

L 158: It would help a reader if the authors could explain how they arrived at those units for the a-coefficient

L 168: A comparison of Eq. 1 and 2 at one or two stations would be interesting. Does the steepness of the fitted rating relation change after the transformation in Eq. 2? If not, please say so. If yes, please document the change.

L 172-179: I find this description a bit hard to follow. Could the authors provide a graphical description/explanation for their computation?

L 185: This comes out of the blue with no explanation. Could the authors please explain WHY bootstrapping? What values are bootstrapped and why?

L 193: ….see Fig. 3 **a**?

L 195 (Fig. 3**b**?)

L 197: clustering around 1? Don't quite agree. Either give a range (between 1 and 1.3) or a value, perhaps 1.2.

L 198: …clustering around 1: don't quite agree with the statement. I'd say: …. Breakpoints of many stations are slightly higher than the geometric mean discharge

L 200: "peaks around 0…" I'd say Fig. 4 indicates that it's > 0. Why not be more specific right away: … peaks near 0.14?

L 205: "The differences…." Sentence sounds off as written. Lowland rivers, by definition, have less relief in their catchments. Reword this sentence such that it does not sound like stating the obvious. The term "topography" is vague, too. Is SSC larger because of a steeper channel gradient at the sampling site or due to steeper gradients in the headwater catchments?

L 208: …are characterized… sounds vague. What about indicating a direct causal relation and say "generate"

L 211: Fig. 7 **a**

L 215: …whether the rating for SSC for these two stations

Consider that an international reader is not aware that the Moselle joins the Rhine in Koblenz. Please reword accordingly.

L 213-216: Those two sentences could be improved. The authors approach the situation with a mindset of: we had a problem and then we solved it. Please try to reword with a standpoint "from above". Also, as written, I would expect a comparison between sites that were sampled weekly and those sampled daily. Instead the reader is shown two sites with seemingly no connection to Koblenz (mind the international reader).

Perhaps something like: A comparison between …. and ….showed that there was not systematic change in bl and bh due to the frequency of sampling.

L 216-218: sentence is off. Place a period after …. And 1.54+- for **the** Moselle). Those values are similar…

Also: the Rhine and the Moselle

L 219: 1000 at each station or at both stations together?

L 220: The authors need to explain the what, where, and why of their bootstrap approach!

L 221: The LOI -measurements…. An introductory sentence is needed here.  The authors just compared the Moselle and the Rhein against other streams.  Now the authors seem to compare between the Moselle and the Rhein.

L 223: …higher LOI values during the summer months… Please show the reader where in Fig. 8 g+h that is to be seen.  Sorry, the color plot shows it.

L 241: "…characterized…"  vague statement.   What about: attributed? Or caused?

L 242: "A positive rating exponent…" It is useful that the authors point to this difference in the exponents of SSC and Qs.  However, this statement appears a bit suddenly. Please provide an introductory sentence.

L. 245: "This implies…" Sentence is poorly worded.  "additional sediment sources": external? Channel bed erosion?

L 247: The authors just switched the discussion from SSC to QS, and I would have expected that the discussion of QS continues, but the authors are switching back to SSC.   Eq. 3 and its explanation is interesting, but it appears that this point is only "squeezed in" and interrupts the thread of arguments.  Please smooth

L. 248: …explained by the increasing…… the reasoning of increasing connectivity and increasing area of water-saturated soils seems to be converted: increase in saturated area causes increase in connectivity.

L 257:  Interestingly, ….An introductory sentence before diving into rating curves from the Elbe and Oder would be useful.

L. 261:  If "reactivity" means that the flow either carries more sed. from its headwaters or pick it up from the channel bed, more explanation than "drier climate" is needed for why the Elbe and Oder do not do so.   Typically, drier areas are considered to have less dense vegetation cover and therefore generate more sediment.  Perhaps the authors might turn to geological conditions:  Sandstone in the Thuringia Forest and glacio-fluvial deposits along Elbe and Oder may be more porous and generate less runoff than the schists in mountains and highlands along the Rhein and in central west Germany.  Perhaps also consider other factors influencing runoff and sediment generation such as land use (percent urban area vs. agriculture) or number of barrages per river mile….?

A detailed discussion of the causes might not be the focus of this paper.  It is ok to say so, but offering an unsupported statement about the effects of a drier climate on SSC and its relation to Q is not satisfactory.

L 264-268: The authors explain that QBr is x times QGM and x times Qavg.  Why is knowing this difference important?

L 274: 1)  "Many of the tributary waterways…"  The authors turn to a new subject.  An introductory sentence is needed, perhaps something pertaining to a assumed relation between reservoir operation, barrages, and a break in the SSC-Q relation.  2) It sounds like the authors are reacting to some instated issue regarding barrages and SSC.  Please bring the reader up to speed on that issue.

L 275-276: What "reservoirs" do the authors refer to?  the channel immediately upstream from a barrage or floodable reservoirs in the floodplain that serve to retain flood waters? What are weir shutters and where are they located? How does opening weir shutters prevent damage to barrages?

L 284:  The authors should elaborate on the "Therefore".  Perhaps something like: Given that the study found this and that, and given that flow management in reservoirs and barrages does not seem to control the SSC-Q relation, …the question remains…

L 285: …at average discharge?  In the analyses, the authors related Qbr to the fraction of Q/QGM. Here, the discussion continues with Qavg.  Why this change?

L 305: Suggest switching the second and first part of the argument: While a positive correlation between SSCtot and Q was observed for most of the year, SSCtot related negatively to Q during the low flow months, indicating the effects of dilution of SSCtot as flows just start to increase and a shift  in the SSC regime…

L. 310: …decreasing trend of bl (Fig. 6b) Should that be Fig 6a?

L 322: Again, what bootstrapping?

L 331 and 360:  …breaks slightly above Q/QGM = 1

L 377: water sampling…. Perhaps: water quality sampling? Or SSC sampling?

**Figures**

Fig. 4: Instead of the four colors are not very distinctive and indistinguishable when viewed in black and white.  I suggest using different line types.

Fig. 6: Please explain the empty circles.

Fig. 6: When viewed in black and white, there is no color distinction between north and south.  Perhaps use a gray scale or patterned circles.

Fig. 7:  The small dots with different colors are not well distinguishable, esp. not in b & w.   Suggest using different symbols.  The x-axis title is not understandable.  Suggest: (%$A_{S>10\%}$) and explaining %A… in the caption text.

Fig. 7: The caption could be better worded:  SR: Relation between rating coefficients (….) and the fraction of the catchment area with hillslopes steeper than 10% (%$A_{S>10\%}$)

Fig. 8:  In caption, replace "line" by "row", and refer to top row and bottom row.

Fig. 9 is overly busy.  Considering that the authors do not discus all plotted statistical information (min, max, outliers), I suggest simplifying the plot to improve its readability and emphasize the plots' main points.  For Fig. 9a I suggest:

- drawing a curve indicating the median values for each month. Surround that curve with a shading the upper and lower boundaries of which indicate the quartile values.

- Do the same for the second site in Fig. 9a but use a distinctly different color scheme.

- Do the same for Fig. 9b.

- Do the same for the inset plot and place as the third panel, the same size as the other two panels between panels a and b.

**Technical comments**
Dear authors: the font size used in this manuscript is annoyingly small!

**SR = suggested rewording**

L 10: …of suspened sediment (omit "the") ….. discharge COLLECTED at 62…

L 17 ff: SR.. likely results from a change of factors controlling suspension of
intrinsic organic matter at low flows to extrinsic sediment supply (including mineral and organic fractions) due to
hillslope erosion at high flows.

L 21: SR:…and facilitates new insights

L 24: SR: Suspended sediment dominates sediment transport

L 30: SR: Dynamics of suspended sediment are strongly influenced by sediment

L 31: SR: Size and density

L 32: SR: Size and density of fine suspended particles in

L 34: SR: Depending on sediment sources…

L 35: SR… topsoil from either hillslopes or

L 36 + 40+41 allochthones   spelling!

L 42: SR: temperatures, light and high….

L 47: SR: even if light, temperature….

L 52-53: no new paragraph needed

L 57: SR: …in turn, affect transport dynamics

L 71: i.e., always followed by a comma

L 71: SR: …as proposed by Reid…

L 75: Q: use italics

L 84: after Asselmann, (2000) SR: as well as combinations of both within one event.

L 84: instead of "characteristics": SR: processes affecting a rating relation in a specific case are well known…

L 90: …inorganic particles in sediment rating curves…

L 92-93: not sure a new paragraph is needed

L 94: … behaviours that are SR: "each controlled by different and independent processes." We test this….

L 114: SR: …waterways is monitored daily using instantaneous water samples (see below) taken manually…

L 116: SR: …in 1965 and has accumulated long-term records

L 120: Frings et al

L 126: SR: The use of coffee…and facilitates measuring SSC at large numbers

L 129 SR: In general, suspended…

L 137-139: SR: Biological fluxes, namely…(Chla) have been monitored since 1997 at two sampling sites located immediately….

L 146: SR: …*LOI*, we segregated …

L 160: …normalized by the … (….) computed for each station according….

L 165: …linked to the response of SSC to changing discharge

L 167: SR: For most gauging stations included in this study, a and b….

L 189: The rating relations for….

L 199: SR: Rating exponents for the….range between

L 200: Sentence gets too long. SR: ….Fig 5). SSC decreases as a function…

L 203: Fig. 6 shows….   More information could be put into that sentence.  SR: Patterns of spatial distribution become apparent (Fig. 6) for the rating coefficients….

L 204: found along the Rhine

L 207: SR: …the fraction of hillslopes steeper than 10% in the contributing catchment area

Or: the fraction of contributing catchment area stepper than 10%

L 213: SR: Considering that water sampling…

L 214: …to daily sampling at the suspended…

L 215: …rating breaks occur at …

L 231-232: SR: Higher Chla-values occur only during moderate flows in spring and summer.  Chla-values in the Rhine peak in April, and in May at the Moselle (Figs. 8 and 9).

L 239: ….Poesen, 2018) and SR:   the presence of this process chain is supported by…

L 252: SR: Our results show a clear trend of increasing bl and bh as the fractions of steep hillslopes with S>10%) increase, thus confirming the expectation.

L 257: SR: Furthermore, our results that show steep rating curves for the Rhine tributaries than the Rhine itself confirm results by Asselmann….

L 259: SR: Assuming similar catchment topographies for a specified percentage of catchment area steeper 10%, the lower SSC generated at high Q in the Elbe and Oder may be attributable to climatic conditions.

L 271: SR: "…significantly to discharge" SR: runoff.  "….but water…" SR: discharge

L 272: SR: …from bl to bh likely reflects a change in factors controlling SSC from ….

L 285:  …show that the contribution of organic suspended matter to total SSC…

L 308: SR: …org. fraction of SSC generally adds a… (or: SSC adds a substantial share to SSCtot year round, the rating…

L 309: SR: For instance, Hardenbicker et al. 2016 reported for the Elbe that LOI and Chla contributions to SSC increased with distance downstream, and this is reflected in the decrease of bl exponents with distance downstream.

L 313:  organic-rich streamflows?    organic-poor

L 327:  The decrease of ….supports

L 346: …load is transported

L 351: SR: in the case of a substantial contribution of the organic SSC to ……. practice of using

L 358: SR: …, but show a distinct

L 359: SSC-Qw     Q was not denoted as Qw previously.

L 361: SR: …likely a result of a change in controlling…of suspended….

L 363: SR: …catchment) sources

L 374: …paper were provided by the suspended….

---

## Referee Comment (RC2) · Anonymous Referee #2 · 31 Mar 2020

Hoffmann et al. present a new conceptual model that allows to distinguish sediment load into organic and inorganic shares. The authors apply this model to an impressive number of gauging stations where manually sediment concentrations are estimated. Basically, the authors apply the classical sediment rating curve, though, extend it to account for varying ratios between organic and inorganic constituents. In general, I see the manuscript by Hoffmann et al. as a relevant contribution and, thus, consider it as worth being published in ESurf.

While reading the manuscript, several concerns and/or suggestions arise:

- The model the authors present is a way to analyze a static system. However, the

authors mention that the model is also applicable to study river dynamics. I think it is important to highlight, what the authors refer to when analyzing the dynamics. As I understand, the authors restrict dynamics in a spatial mode, i.e. intrinsic vs. extrinsic. What the author don't study, and I think this is important to mention, is the temporal dynamics. As the authors state in the introduction, temporal dynamics may be analyzed using hysteresis loops (among other). Maybe it is too much additional work and maybe beyond the scope of this manuscript: Did the authors looked on the hysteresis loops, too? I think this is important, at least, to be discussed.

- The section of the three methods applied to quantify the scale-breaks of suspended sediment is a bit unclear. I am convinced that better explaining the three distinct methods, eventually doing a bit more math, would improve the manuscript. For example, I cannot see how the authors defined the subsets used in the second method, i.e. how do the authors construct the "sequences" of discharge $Q_i$? I am also curious why the authors did not use a change-point detection algorithm and applied a piecewise regression to a lower and higher flow regime. I am not saying that the approach chosen by the authors is "wrong", yet I was just interested in more details on the methods chosen.

- The authors considered the geometric mean in their study. Later in the manuscript, they state, however, that the simple average is $\sim$0.8 x the geometric mean (L 269). I am wondering why the authors did not chose a simple average from the very beginning?

- Regarding the sampling routine, I was wondering if the same sampling protocol has been applied for both the daily and weekly measurements? Did the sampling involve also depth-integration?

- The authors explain possible interpretations of the coefficients. Yet, the part around line 159 (MTb L-(1+3b)) is not clearly written. Maybe the authors can provide some better explanation to follow their reasoning.

- L 178: Therefore, Q values were classified into equally spaced classes at a log-scale. How many classes exactly?

- L 150: "Chla was used as a proxy. . . for biomass dynamics" What do the authors refer to here exactly when mentioning dynamics? Better to use simply load?

- L8: major relevance for sustainable sediment management. What is that exactly and maybe I missed it, but where do the authors consider this in their manuscript?

- L 44: "Water flow velocities regulate the water residence times, which in turn affect the time for phytoplankton growth in river systems. Low flow conditions with increased residence times provide favourable conditions for phytoplankton growth or even blooms. In contrast, short residence times can strongly reduce the share of autochthonous biomass in suspended sediments, even if light availability, temperature and nutrient levels are not limited (Fischer, 2015; Quiel et al., 2011)." This argument is not completely clear to me. I see the time restrictions for phytoplankton growth given a fast draining river. However, it depends on where you sample, I guess, too. Given high flow velocities, I assume that the concentration of phytoplankton is indeed relatively low in the water column. However, as load is the product of concentration times discharge, the overall phytoplankton load may be high, too. I am not a biologist. Maybe the authors can better explain their thoughts on that and how this may affect the results and findings they present here.

- The authors used coffee filter and stated that the pore diameters of 0.7 to 1 $\mu$m. How was this number determined?

- L 109: Specific discharge. I assume that this is well known to most of the readers. Regardless, I think it would be good to define it here. The same is true for "long-term discharge weighted averages of SSC". Please define this, too.

- L 143 ff. The way LOI is explained here is not completely clear. Based on the context, LOI is here defined as the fraction of the total load, i.e. 0-1. However, the authors also write that "The organic component was combusted at 500°C for 1 hour to estimate the LOI of the suspended matter." This sentence implies a mass involved and, thus, units. Please clarify. See also L 291: "Here we use LOI as a measure of the organic

fraction of the total suspended solids." Maybe the latter sentence can be moved into the methods section?

- The authors applied the t-test to test the rating coefficients. Are the samples normally distributed and all other requirements met? If not, the t-test is not applicable.

- L 194: "For 52 out of 62 stations, SSC - Q rating curves show a distinct break in scaling relation (for examples see Fig. 3) with similar values for Qb estimated from three different approaches (Tab. 2)." Is there any spatial pattern in terms of signal propagation along nested catchments? This would be an interesting finding.

- L 310: "At stations where the organic fraction of the SSC adds a substantial share to the total SSC,..." What is substantial?

- L 207: "This control is highlighted in Fig. 7, which plots bh with respect to the fraction of the contributing catchment area that is steeper than 10% slope gradient. Catchments with a higher fraction of steep slopes are characterized be higher bh -values." While this finding is somehow expected, I was wondering how the authors decided to choose the 10% value? Why didn't the authors consider all percentiles, i.e. involving the entire topography? 10% sounds a bit arbitrary to me.

- L 220: "However, the lower number of measurements at the LOI-stations (approx. 1000 at both stations) resulted in a larger uncertainty of the parameter estimation ($\Delta$bl and $\Delta$bh) from the bootstrap regression". Can the authors somehow quantify the involved uncertainties?

- L 226: "resulting in rating exponents b of $-0.51 \pm 0.03$ and $-0.47 \pm 0.01$ , and a -coefficients of $0.202\pm0.003$ and $0.319\pm0.006$ for the Rhine and the Moselle, respectively". Please include b here; It makes the reading a lot easier.

- L 261: "The dry continental climate in the Elbe and Oder catchments likely reduces the reactivity of the river systems, requiring larger increases of rain and discharge to increase the specific sediment supply in these basins compared to basins with
higher/more frequent precipitation in the western part of Germany." This is a reasonable interpretation. Yet, can the authors provide a reference? Or can the authors estimate catchment-averaged rainfall and relate this to the sediment fluxes observed?

- L 274: "Thus, the transition from bl to bh is likely to be a result of a change of controlling factors of the suspended sediment from intrinsic (within the river system) to extrinsic (outside the river channel but within the catchment) factors." Well, this is just a personal suggestion: I suggest do avoid intrinsic and extrinsic in this case here: It is a hydrological system, though. Given the catchment scale used here, intrinsic suggest within the catchment and extrinsic from outside the catchment. However, I leave this up to the authors and editors.

- L 353: "In the case of substantial share of the organic SSC to the total SSC, our results suggest that the common practice using a continuous sediment rating results in large errors that can be reduced applying rating relationship including scale breaks." Well, does this really matter if organic transport shares are only important during low flows? I would assume that temporal changes in the sediment rating (hysteresis) might be equally important or even more important. In fact, this study shows that larger fraction of organic matter remains unconsidered during low flows only.

---

## Author Comment (AC1) · 20 Apr 2020

First of all, we thank Kristin Bunte and the anonymous reviewer for their constructive criticism and for taking the time to share their insightful suggestions and comments. Both reviewers do not have a general critic on the manuscript, but made many detailed suggestions how to improve it. Thus, we address their comments in the revised version of the manuscript (including track changes option) and reply to their detailed comments step by step below. We have the feeling that the manuscript greatly improved thanks to the efforts of the reviewers.

Both reviewers raised their concern regarding the description of the methods applied

to compute the breakpoint between the low-flow and high-flow. We critically reviewed the methods section with a special emphasis on bootstrapping and sincerely hope that these changes make our manuscript more to the point and easier to follow. Also, we gave more information on why the geometric mean is used for normalization.

Furthermore, we point out that in reaction to the valid criticism raised by Kristin Bunte, we changed large parts of section 4.1 to ensure a better connection between the aspects discussed therein.

We fully agree with reviewer #2 that we used a static approach to analyze the sediment rating. We rephrased the manuscript to clarify this confusion.

All other detailed comments will be addressed in the revised manuscript. A detailed reply can be found in the attached file.

Kind regards

On behalf of all co-authors Thomas Hoffmann

Please also note the supplement to this comment:
https://www.earth-surf-dynam-discuss.net/esurf-2020-3/esurf-2020-3-AC1-supplement.pdf

**Supplement:**

**Revision of the manuscript on "Scale-breaks of suspended sediment rating in large rivers in Germany induced by organic matter" by Thomas O. Hoffmann et al.**

First of all, we thank Kristin Bunte and the anonymous reviewer for their constructive criticism and for taking the time to share their insightful suggestions and comments. Both reviewers do not have a general critic on the manuscript, but made many detailed suggestions how to improve it. Thus, we address their comments in the revised version of the manuscript (including track changes option) and reply to their detailed comments step by step below. For your convenience, we colored our replies to each comment of the reviewers in green letters.

**Referee #1: Kristin Bunte**

The manuscript presents an interesting study and is well worth publishing in e-surf. Teasing apart the different sedimentary constituents (mineral, organic) that determine rating relation between suspended sediment and discharge is an important contribution. The authors quantify the different components and their change with discharge and provide a model that appears to be applicable to streams in Germany.
The description of the three methods used to compute the breakpoint flow is a bit unclear, a bit more help is needed to enable the reader the follow all steps. More of a problem is that the authors repeatedly refer to a bootstrapping approach, but never explain why bootstrapping is undertaken, what data are used, and what the purpose of the bootstrapping approach is in the first place. This needs to be revised. The manuscript also needs clarifications all over. Most of the issues are minor and can be easily addressed. I have provided a lot of suggestions for the authors when tackling those issues.
The title reflects the topic of the study, and the abstract summarizes the study effectively. The manuscript is generally well structured. An exception is the discussion Section 4.1 which is weaker than the other chapters. Items in Section 4.1 are discussed one after the other without connection, without introduction, and without a stated course of argument in the beginning.
Use of the English language is fair. The manuscript needs editorial improvement, the majority of which are minor corrections that can be easily addressed.
Several of the figures need improvement; Again, nothing serious, but revision would help to improve the manuscript quality.
In all, my evaluation of the manuscript is: publish with minor revisions, of which there are a lot, but most of them can be addressed in a straightforward way.

We thank Kristin for her very helpful and detailed comments on our manuscript. Before addressing the detailed comments below, we point out that in reaction to the valid criticism raised here, we changed large parts of section 4.1 to ensure a better connection between the aspects discussed therein. Furthermore, we critically reviewed the methods section with

a special emphasis on bootstrapping and sincerely hope that these changes make our manuscript more to the point and easier to follow.

**Comments requesting clarification:**

L. 15: … identify discharge depended process regimes of suspended matter…. Please define better. Too short - jargon. → rephrased to '…discharge dependent controls of suspended matter.' to avoid jargon.

L 21: …into the first order control of discharge dynamics of suspended sediments. Sounds like jargon. Please start more clearly. → rephrased to '…into the first order control of discharge on the quality and quantity of suspended sediments.' to avoid jargon.

L 63: Please define more clearly which Q is meant. Instant Q? Mean daily Q? → basically, a whole range of "Q" can be used depending on the approach and available data. We added a phrase explaining this: … "Rating curves plot $SSC$ as a function of water discharge $Q$, while the temporal aggregation (or resolution) depends on the approach and available data and ranges from 15 minutes to annual averages."

L 77: The authors get too involved in describing their plot. I think I know what the authors mean, but I think they could describe this better. SR: A first look at the plot of measured values of SSC vs Q in which data are not segregates by time or processes controlling SSC exhibits strong scatter. → we rephrased the sentence.

L 80: The authors might add that changing water supply or flow hydraulics could also be at play. → we added flow hydraulics because changing water supply is associated with changing Q and changing water sources.

L 86: The authors should add as a 4th item to the list the effects of organic material on SSC that were introduced earlier. Doing so would also give the authors the opportunity to create a connection to the thought starting in line 86 which, as written, appears out of context. SR: Recently, we have learned more about the effects of organic material on SSC, but so far, "Most monitoring studies… → We did not add a 4th item but used the SR to build a bridge between both paragraphs.

L101: Please clarify: Do the "daily" discharge measurements reflect once-a-day instantaneous measurements or are daily averages computed from or continuous measurements of Q and SSC? → We agree that more detailed infos on the measurements is needed, but not at this place. Thus, we deleted the word daily in this line and extended the description in chapter 2.2.

L 116: Here, we selected….Please explain the reasoning for the selection → done, we added more information.

L 123: …SSC was given… do you mean "computed" → yes, rephrased

L 128: (e.g., medium and finer clays?) → rephrased

L 158: It would help a reader if the authors could explain how they arrived at those units for the a-coefficient → done, given the additional information the reader should be able to reproduce the dimension analysis to highlight the dependency between a and b.

L 168: A comparison of Eq. 1 and 2 at one or two stations would be interesting. Does the steepness of the fitted rating relation change after the transformation in Eq. 2? If not, please say so. If yes, please document the change. → the steepness of the fitted relation does not change, we provide more information in the text accordingly.

L 172-179: I find this description a bit hard to follow. Could the authors provide a graphical description/explanation for their computation? → This part was strongly restructured. We hope that it is easy easier to follow now.

L 185: This comes out of the blue with no explanation. Could the authors please explain WHY bootstrapping? What values are bootstrapped and why? → We strongly rephrased this paragraph to give more insights into the WHY.

L 193: ….see Fig. 3 **a**? → done

L 195 (Fig. 3**b**?) → done

L 197: clustering around 1? Don't quite agree. Either give a range (between 1 and 1.3) or a value, perhaps 1.2. → rephrased to give the range of the first and third quantile.

L 198: …clustering around 1: don't quite agree with the statement. I'd say: …. Breakpoints of many stations are slightly higher than the geometric mean discharge → adopted accordingly

L 200: "peaks around 0..." I'd say Fig. 4 indicates that it's > 0. Why not be more specific right away: … peaks near 0.14? → done

L 205: "The differences…." Sentence sounds off as written. Lowland rivers, by definition, have less relief in their catchments. Reword this sentence such that it does not sound like stating the obvious. The term "topography" is vague, too. Is SSC larger because of a steeper channel gradient at the sampling site or due to steeper gradients in the headwater catchments? → we deleted the sentence and rephrased the paragraph!

L 208: …are characterized… sounds vague. What about indicating a direct causal relation and say "generate" → done

L 211: Fig. 7 **a** → done

L 215: …whether the rating for SSC for these two stations → done

Consider that an international reader is not aware that the Moselle joins the Rhine in Koblenz. Please reword accordingly. → it was already state in chapter 2.2. that both rivers join in Koblenz. However, we repeated it here.

L 213-216: Those two sentences could be improved. The authors approach the situation with a mindset of: we had a problem and then we solved it. Please try to reword with a standpoint "from above". Also, as written, I would expect a comparison between sites that were sampled weekly and those sampled daily. Instead the reader is shown two sites with seemingly no connection to Koblenz (mind the international reader).

Perhaps something like: A comparison between …. and ….showed that there was not systematic change in bl and bh due to the frequency of sampling. → rephrased as suggested

L 216-218: sentence is off. Place a period after …. And 1.54+- for **the** Moselle). Those values are similar…

Also: the Rhine and the Moselle → rephrased

L 219: 1000 at each station or at both stations together? → rephrased

L 220: The authors need to explain the what, where, and why of their bootstrap approach! → now explained in detail earlier in chapter 2.

L 221: The LOI -measurements…. An introductory sentence is needed here. The authors just compared the Moselle and the Rhein against other streams. Now the authors seem to compare between the Moselle and the Rhein. → rephrased

L 223: …higher LOI values during the summer months… Please show the reader where in Fig. 8 g+h that is to be seen. Sorry, the color plot shows it. → ok, nothing to correct ;-)

L 241: "…characterized…" vague statement. What about: attributed? Or caused? → rephrased to 'attributed'

L 242: "A positive rating exponent…" It is useful that the authors point to this difference in the exponents of SSC and Qs. However, this statement appears a bit suddenly. Please provide an introductory sentence. → this part is rephrased to avoid breaks in the argumentation.

L. 245: "This implies…" Sentence is poorly worded. "additional sediment sources": external? Channel bed erosion? → reworded, additional sources are not specific at this stage, however we get back to this term later in the paragraph, indicating that the additional sources are external.

L 247: The authors just switched the discussion from SSC to QS, and I would have expected that the discussion of QS continues, but the authors are switching back to SSC. Eq. 3 and its explanation is interesting, but it appears that this point is only "squeezed in" and interrupts the thread of arguments. Please smooth → the switch is needed to argue that "additional sediment sources" need to be mobilized, this does not follow from SSC alone.

L. 248: …explained by the increasing…… the reasoning of increasing connectivity and increasing area of water-saturated soils seems to be converted: increase in saturated area causes increase in connectivity. → rephrased

L 257: Interestingly, ….An introductory sentence before diving into rating curves from the Elbe and Oder would be useful. → done

L. 261: If "reactivity" means that the flow either carries more sed. from its headwaters or pick it up from the channel bed, more explanation than "drier climate" is needed for why the Elbe and Oder do not do so. Typically, drier areas are considered to have less dense vegetation cover and therefore generate more sediment. Perhaps the authors might turn to geological conditions: Sandstone in the Thuringia Forest and glacio-fluvial deposits along Elbe and Oder may be more porous and generate less runoff than the schists in mountains and highlands along the Rhein and in central west Germany. Perhaps also consider other factors influencing runoff and sediment generation such as land use (percent urban area vs. agriculture) or number of barrages per river mile….? A detailed discussion of the causes might not be the focus of this paper. It is ok to say so, but offering an unsupported statement about the effects of a drier climate on SSC and its relation to Q is not satisfactory. → We are thankful for this suggestion on the importance of soil-saturation. We added the differences of soil porosity, after highlighting the importance of soil-moisture as a controlling factor of hillslope connectivity.

L 264-268: The authors explain that QBr is x times QGM and x times Qavg. Why is knowing this difference important? → We added the median discharge as well. We make the link to Qavg and Qmed since this is much more familiar to geoscientist and hydrologists than the geometric mean. Furthermore the median discharge allows to link the duration a river spends in the low flow and high flow regime.

L 274: 1) "Many of the tributary waterways…" The authors turn to a new subject. An introductory sentence is needed, perhaps something pertaining to a assumed relation between reservoir operation, barrages, and a break in the SSC-Q relation. 2) It sounds like the authors are reacting to some instated issue regarding barrages and SSC. Please bring the reader up to speed on that issue. → Here we just intended to support evidence that reservoir operation is not the dominant control, but that the issue is more complicated. We rephrased this paragraph slightly to build a bridge to the preceding paragraph.

L 275-276: What "reservoirs" do the authors refer to? the channel immediately upstream from a barrage or floodable reservoirs in the floodplain that serve to retain flood waters? What are weir shutters and where are they located? How does opening weir shutters prevent damage to barrages? → We basically state 'Reservoirs upstream of the barrages…'. This should answer the question. However, to avoid confusion, we rephrased the end of this paragraph from 'reservoir management' to 'operation of barrages'. Explaining the engineering details of barrages is certainly not the aim of this paper. For details the reader is referred to Hoffmann et al. (2017).

L 284: The authors should elaborate on the "Therefore". Perhaps something like: Given that the study found this and that, and given that flow management in reservoirs and barrages does not seem to control the SSC-Q relation, …the question remains… → we rephrased this part as suggested and hope to improve the line of argumentation.

L 285: …at average discharge? In the analyses, the authors related $Q_{br}$ to the fraction of Q/QGM. Here, the discussion continues with $Q_{avg}$. Why this change? → because most people are not familiar with Q_GM and Q_avg is the more common used metric.

L 305: Suggest switching the second and first part of the argument: While a positive correlation between $SSC_{tot}$ and Q was observed for most of the year, $SSC_{tot}$ related negatively to Q during the low flow months, indicating the effects of dilution of $SSC_{tot}$ as flows just start to increase and a shift in the SSC regime… → rephrased as suggested

L. 310: …decreasing trend of bl (Fig. 6b) Should that be Fig 6a? → correct! We change it to Fig 6a

L 322: Again, what bootstrapping? → bootstrapping is a standard procedure to estimate confidence intervals of regression coefficients, as explained earlier!

L 331 and 360: …breaks slightly above Q/QGM = 1 → done, rephrased as suggested

L 377: water sampling…. Perhaps: water quality sampling? Or SSC sampling? → rephrased

**Figures**

Fig. 4: Instead of the four colors are not very distinctive and indistinguishable when viewed in black and white. I suggest using different line types. → done

Fig. 6: Please explain the empty circles. → done

Fig. 6: When viewed in black and white, there is no color distinction between north and south. Perhaps use a gray scale or patterned circles. → Translating the colors to gray scale will lose much of the visual information. The colored version will be available open access online! Thus, we intend not to change the colors.

Fig. 7: The small dots with different colors are not well distinguishable, esp. not in b & w. Suggest using different symbols. The x-axis title is not understandable. Suggest: (%A S>10%) and explaining %A… in the caption text. → points have now different color AND form. Label of x-axis was renamed as suggested

Fig. 7: The caption could be better worded: SR: Relation between rating coefficients (….) and the fraction of the catchment area with hillslopes steeper than 10% (%A S>10%) → rephrased

Fig. 8: In caption, replace "line" by "row", and refer to top row and bottom row. → done

Fig. 9 is overly busy. Considering that the authors do not discus all plotted statistical information (min, max, outliers), I suggest simplifying the plot to improve its readability and emphasize the plots' main points. For Fig. 9a I suggest:
- drawing a curve indicating the median values for each month. Surround that curve with a shading the upper and lower boundaries of which indicate the quartile values.
- Do the same for the second site in Fig. 9a but use a distinctly different color scheme.
- Do the same for Fig. 9b.
- Do the same for the inset plot and place as the third panel, the same size as the other two panels between panels a and b. → It is certainly true that Fig. 9 is quite busy and that not all of the information contained is discussed in detail in the manuscript. However, we are confident that some readers are interested in the full spread of the data. Reducing the data to median and quartile values disguises the extremes and means a loss of information that we want to keep, be it at the expense of readability.

**Technical comments**

Dear authors: the font size used in this manuscript is annoyingly small!

We are very sorry for that but have to pass this one to Copernicus and their journals. We used the downloadable Word-Template that uses Times New Roman in 10 pt.

**SR = suggested rewording**

L 10: …of suspened sediment (omit "the") ….. discharge COLLECTED at 62… → done

L 17 ff: SR.. likely results from a change of factors controlling suspension of intrinsic organic matter at low flows to extrinsic sediment supply (including mineral and organic fractions) due to hillslope erosion at high flows. → done

L 21: SR:…and facilitates new insights → done

L 24: SR: Suspended sediment dominates sediment transport → done

L 30: SR: Dynamics of suspended sediment are strongly influenced by sediment → rephrased differently

L 31: SR: Size and density → done

L 32: SR: Size and density of fine suspended particles in → done

L 34: SR: Depending on sediment sources… → done

L 35: SR… topsoil from either hillslopes or → done

L 36 + 40+41 allochthones spelling! → we used the spelling as given

L 42: SR: temperatures, light and high…. → rephrased

L 47: SR: even if light, temperature…. → rephrased

L 52-53: no new paragraph needed → removed the new line

L 57: SR: …in turn, affect transport dynamics → done

L 71: i.e., always followed by a comma → done

L 71: SR: …as proposed by Reid… → done

L 75: Q: use italics → done

L 84: after Asselmann, (2000) SR: as well as combinations of both within one event. → done

L 84: instead of "characteristics": SR: processes affecting a rating relation in a specific case are well known…

L 90: …inorganic particles in sediment rating curves… → done

L 92-93: not sure a new paragraph is needed → we would like to keep it here since we want the hypothesis to start with a new paragraph!

L 94: … behaviours that are SR: "each controlled by different and independent processes." We test this…. → d one

L 114: SR: …waterways is monitored daily using instantaneous water samples (see below) taken manually… → done

L 116: SR: …in 1965 and has accumulated long-term records

L 120: Frings et al → done

L 126: SR: The use of coffee…and facilitates measuring SSC at large numbers → done

L 129 SR: In general, suspended… → done

L 137-139: SR: Biological fluxes, namely…(Chla) have been monitored since 1997 at two sampling sites located immediately…. → done

L 146: SR: …*LOI*, we segregated … → done

L 160: …normalized by the … (….) computed for each station according…. → done

L 165: …linked to the response of SSC to changing discharge → done

L 167: SR: For most gauging stations included in this study, a and b…. → done

L 189: The rating relations for…. → done
L 199: SR: Rating exponents for the….range between → done
L 200: Sentence gets too long. SR: ….Fig 5). SSC decreases as a function… → done
L 203: Fig. 6 shows…. More information could be put into that sentence. SR: Patterns of spatial distribution become apparent (Fig. 6) for the rating coefficients…. → done
L 204: found along the Rhine → done
L 207: SR: …the fraction of hillslopes steeper than 10% in the contributing catchment area
Or: the fraction of contributing catchment area stepper than 10% → done
L 213: SR: Considering that water sampling… → this paragraph has been strongly rephrased
L 214: …to daily sampling at the suspended… → dito
L 215: …rating breaks occur at … → done
L 231-232: SR: Higher Chla-values occur only during moderate flows in spring and summer. Chla-values in the Rhine peak in April, and in May at the Moselle (Figs. 8 and 9). → done
L 239: ….Poesen, 2018) and SR: the presence of this process chain is supported by… → done
L 252: SR: Our results show a clear trend of increasing bl and bh as the fractions of steep hillslopes with S>10%) increase, thus confirming the expectation. → done
L 257: SR: Furthermore, our results that show steep rating curves for the Rhine tributaries than the Rhine itself confirm results by Asselmann…. → done
L 259: SR: Assuming similar catchment topographies for a specified percentage of catchment area steeper 10%, the lower SSC generated at high Q in the Elbe and Oder may be attributable to climatic conditions. → partially rephrased
L 271: SR: "…significantly to discharge" SR: runoff. "….but water…" SR: discharge → done
L 272: SR: …from bl to bh likely reflects a change in factors controlling SSC from …. → done
L 285: …show that the contribution of organic suspended matter to total SSC… → done
L 308: SR: …org. fraction of SSC generally adds a… (or: SSC adds a substantial share to SSCtot year round, the rating… → done
L 309: SR: For instance, Hardenbicker et al. 2016 reported for the Elbe that LOI and Chla contributions to SSC increased with distance downstream, and this is reflected in the decrease of bl exponents with distance downstream. → done
L 313: organic-rich streamflows? organic-poor → organic rich!
L 327: The decrease of ….supports → done
L 346: …load is transported → done
L 351: SR: in the case of a substantial contribution of the organic SSC to ……. practice of using → done
L 358: SR: …, but show a distinct → done
L 359: SSC-Qw Q was not denoted as Qw previously. → changed
L 361: SR: …likely a result of a change in controlling…of suspended…. → done
L 363: SR: …catchment) sources → done
L 374: …paper were provided by the suspended…. → done

**Anonymous Referee #2**

Hoffmann et al. present a new conceptual model that allows to distinguish sediment load into organic and inorganic shares. The authors apply this model to an impressive number of gauging stations where manually sediment concentrations are estimated. Basically, the

authors apply the classical sediment rating curve, though, extend it to account for varying ratios between organic and inorganic constituents. In general, I see the manuscript by Hoffmann et al. as a relevant contribution and, thus, consider it as worth being published in ESurf. → We are thankful for the many good suggestions of reviewer #2. Again, we commented on each suggestion (see text in green letters) and revised the text in most of the cases as suggested.

While reading the manuscript, several concerns and/or suggestions arise:

- The model the authors present is a way to **analyze a static system**. However, the authors mention that the model is also applicable to study river dynamics. I think it is important to highlight, what the authors refer to when analyzing the dynamics. As I understand, the authors restrict dynamics in a spatial mode, i.e. intrinsic vs. extrinsic. What the author don't study, and I think this is important to mention, is the temporal dynamics. As the authors state in the introduction, temporal dynamics may be analyzed using hysteresis loops (among other). Maybe it is too much additional work and maybe beyond the scope of this manuscript: Did the authors looked on the hysteresis loops, too? I think this is important, at least, to be discussed.
  ⇒ We agree with the reviewer that the term 'dynamics' is misleading in our context, since we do not analyze the changes of the rating behavior, nor consider hysteresis curves (which is far beyond the scope of this study). Therefore, we changed the wording (mainly in the introduction) to avoid the expectation that our concept is based on a dynamic (time-variable) approach. We hope that this matches the concerns raised by the reviewer

- The section of the three methods applied to quantify the scale-breaks of suspended sediment is a bit unclear. I am convinced that better explaining the three distinct methods, eventually doing a bit more math, would improve the manuscript. For example, I cannot see how the authors defined the subsets used in the second method, i.e. how do the authors construct the "sequences" of discharge Qi? I am also curious why the authors did not use a change-point detection algorithm and applied a piecewise regression to a lower and higher flow regime. I am not saying that the approach chosen by the authors is "wrong", yet I was just interested in more details on the methods chosen.
  ⇒ Basically, we did a piecewise regression as suggested by the reviewer to detect the change-point between the low and high flow. We hope that the strongly rephrased paragraph avoids any potential confusion.

- The authors considered the geometric mean in their study. Later in the manuscript, they state, however, that the simple average is ~0.8 x the geometric mean (L 269). I am wondering why the authors did not chose a simple average from the very beginning?
  ⇒ If the steepness of the regression line in a scatter plot changes without a change in the mean of the y-values (e.g. the SSC-values), then the lines circulate around the geometric mean of the x-values (see Warrick, 2015). That is the reason why we used the geometric mean to normalize the values and therefore to achieve independence of the regression coefficients a and b. However, we related the $Q_{gm}$ to the $Q_{median}$ and $Q_{avg}$ later on, since these are much more familiar to most geoscientists and hydrologists. Additionally, the median allows to refer to the time the river systems spend in the high flow and low flow regime (50/50).

- Regarding the sampling routine, I was wondering if the same sampling protocol has been applied for both the daily and weekly measurements? Did the sampling involve also depth-integration?
  $\Rightarrow$ We added some more information regarding the water sampling to clarify the questions of reviewer. We clearly state now, that water sampling was limited to the top 50cm of the water surface. Thus, no depth-integrated sampling was applied.
- The authors explain possible interpretations of the coefficients. Yet, the part around line 159 (MTb L-(1+3b) is not clearly written. Maybe the authors can provide some better explanation to follow their reasoning. → We gave additional information and explanation regarding the units.
- L 178: Therefore, Q values were classified into equally spaced classes at a log-scale. How many classes exactly? → the number of classes was variable (depending on the Q-range); however the width of the classes was constant, we added this info in the text.
- L 150: "Chla was used as a proxy ... for biomass dynamics" What do the authors refer to here exactly when mentioning dynamics? Better to use simply load? → we removed the word dynamics and simply state that chla is as proxy of phytoplankton biomass!
- L8: major relevance for sustainable sediment management. What is that exactly and maybe I missed it, but where do the authors consider this in their manuscript? → Indeed, we did not discuss the implications of our results for sustainable sediment management. This introductory sentence aims to motivate the reader to show the general implications why a good understanding of the processes of suspended sediment transport are needed, without aiming to discuss these implications in detail.
- L 44: "Water flow velocities regulate the water residence times, which in turn affect the time for phytoplankton growth in river systems. Low flow conditions with increased residence times provide favorable conditions for phytoplankton growth or even blooms. In contrast, short residence times can strongly reduce the share of autochthonous biomass in suspended sediments, even if light availability, temperature and nutrient levels are not limited (Fischer, 2015; Quiel et al., 2011)." This argument is not completely clear to me. I see the time restrictions for phytoplankton growth given a fast draining river. However, it depends on where you sample, I guess, too. Given high flow velocities, I assume that the concentration of phytoplankton is indeed relatively low in the water column. However, as load is the product of concentration times discharge, the overall phytoplankton load may be high, too. I am not a biologist. Maybe the authors can better explain their thoughts on that and how this may affect the results and findings they present here. → Here we consider mainly the control of (mineral and organic) suspended sediment concentrations and highlight the fact that autochthonous organic matter has a contrasting relationship to discharge compare to allochthonous susp. matter. The effect on the load is discussed in chapter 4.3. We added a sentence to highlight this difference and to show that the focus here is rather on concentration than on load.
- The authors used coffee filter and stated that the pore diameters of 0.7 to 1 μm. How was this number determined? → This was done in an earlier study of our sediment lab (published in German only). This study compared grain size analysis of suspended sediment before and after filtering. We rephrased the relevant sentence to hint to this approach, but we do not intend to show the results here.
- L 109: Specific discharge. I assume that this is well known to most of the readers. Regardless, I think it would be good to define it here. The same is true for "long-term discharge weighted averages of SSC". Please define this, too. → specific discharge is not

defined. We added a line in chapter 2.2. to inform about the calculation of discharge
weighted SSC.

- L 143 ff. The way LOI is explained here is not completely clear. Based on the context, LOI
  is here defined as the fraction of the total load, i.e. 0-1. However, the authors also write
  that "The organic component was combusted at 500∘C for 1 hour to estimate the LOI of
  the suspended matter." This sentence implies a mass involved and, thus, units. Please
  clarify. See also L 291: "Here we use LOI as a measure of the organic fraction of the total
  suspended solids." Maybe the latter sentence can be moved into the methods section?
  → we added a sentence that clearly defines that LOI is give as a fraction of the total SSC:
  'In our study LOI is give as the ratio of organic suspended matter to the total SSC.'

- The authors applied the t-test to test the rating coefficients. Are the samples normally
  distributed and all other requirements met? If not, the t-test is not applicable. → we
  check for normal distributions. We added some information to the boostrapping, stating
  that distributions were normal distributed.

- L 194: "For 52 out of 62 stations, SSC - Q rating curves show a distinct break in scaling
  relation (for examples see Fig. 3) with similar values for Qb estimated from three
  different approaches (Tab. 2)." Is there any spatial pattern in terms of signal propagation
  along nested catchments? This would be an interesting finding. → We did not check for
  the spatial pattern pf the break point. Given the rather narrow distributions of Q_br and
  the uncertainty related to the estimation of Q_br, we doubt that the differences
  between stations along a single river channel provides meaningful information. This
  could however be part of a future paper ;-)

- L 310: "At stations where the organic fraction of the SSC adds a substantial share to the
  total SSC,..." What is substantial? → we added some more information regarding the
  Moselle where this share is roughly 60% at low flows.

- L 207: "This control is highlighted in Fig. 7, which plots bh with respect to the fraction of
  the contributing catchment area that is steeper than 10% slope gradient. Catchments
  with a higher fraction of steep slopes are characterized be higher bh -values." While this
  finding is somehow expected, I was wondering how the authors decided to choose the
  10% value? Why didn't the authors consider all percentiles, i.e. involving the entire
  topography? 10% sounds a bit arbitrary to me. → The reviewer is correct. 10 % is
  arbitrarily chosen. However, the result would not change much if another threshold is
  chosen.

- L 220: "However, the lower number of measurements at the LOI-stations (approx. 1000
  at both stations) resulted in a larger uncertainty of the parameter estimation (Δbl and
  Δbh) from the bootstrap regression". Can the authors somehow quantify the involved
  uncertainties? → the larger uncertainty results from the larger standard deviation of the
  distributions of the estimated parameters, as denoted by the terms in the brackets. To
  clarify this, we changed the word 'uncertainty' with 'standard deviation'

- L 226: "resulting in rating exponents b of −0.51 ± 0.03 and −0.47 ± 0.01 , and a -
  coefficients of 0.202±0.003 and 0.319±0.006 for the Rhine and the Moselle, respec-
  tively". Please include b here; It makes the reading a lot easier. → we included b here as
  suggested

- L 261: "The dry continental climate in the Elbe and Oder catchments likely reduces the
  reactivity of the river systems, requiring larger increases of rain and discharge to increase
  the specific sediment supply in these basins compared to basins with higher/more
  frequent precipitation in the western part of Germany." This is a reason- able
  interpretation. Yet, can the authors provide a reference? Or can the authors estimate

catchment-averaged rainfall and relate this to the sediment fluxes observed? → Due to the suggestions of Kristin Bunte, we revised this discussion with a stronger focus on soil moisture. Thus, we are not sure if simply annual rainfalls will provide much more information. Certainly, we are not able to give numbers for (antecedent) soil moisture; this is far beyond the scope of the paper.

- L 274: "Thus, the transition from bl to bh is likely to be a result of a change of controlling factors of the suspended sediment from intrinsic (within the river system) to extrinsic (outside the river channel but within the catchment) factors." Well, this is just a personal suggestion: I suggest do avoid intrinsic and extrinsic in this case here: It is a hydrological system, though. Given the catchment scale used here, intrinsic suggest within the catchment and extrinsic from outside the catchment. However, I leave this up to the authors and editors. → The reviewer is correct if we would analyze the holistic sediment budget of a river catchment. However, our main focus here is the river channel. To avoid the complications due to the potentially different views regarding these terms, we defined the meaning of intrinsic and extrinsic within the brackets. We have the feeling that this should avoid any confusion.

- L 353: "In the case of substantial share of the organic SSC to the total SSC, our results suggest that the common practice using a continuous sediment rating results in large errors that can be reduced applying rating relationship including scale breaks." Well, does this really matter if organic transport shares are only important during low flows? I would assume that temporal changes in the sediment rating (hysteresis) might be equally important or even more important. In fact, this study shows that larger fraction of organic matter remains unconsidered during low flows only. → Again, we focus our discussion on the suspended sediment concentration. Indeed, geomorphologist are more concerned about loads and fluxes and the effect of the uncertainty of loads is much smaller than that on concentrations. However, from a biologist point of view, concentrations of organic matter are of great importance for biological processes. Finally, increased suspended concentrations (due to the high organic matter content) at low flow effect the global regression, leading to underestimates of the regressed SSC compared to measured values). We explained these effects on loads by adding an additional sentence at the end of this paragraph.

---

## Editor Comment (EC1) · Robert Hilton (Editor) · 28 Apr 2020

Dear Authors, The discussion phase of the peer review process has now closed. The reviewers were positive about your manuscript and its contribution to ESurf. Thank you for your replies to the reviewers comments. These are generally thorough and indicate that a revised version of the manuscript can address these comments. However, I have completed my own detailed review, and this has flagged a few additional points to address.

Here, I provide further feedback based on my own reading, and indicate points which perhaps could be further clarified in response to the reviewers.

[Figure]

Referee 2: Their first comment about temporal changes – I didn't see much included on this point in the revised version. I think it's a fair point to make (indeed there are some discussions about temporal changes in the manuscript). It seems sensible to include some more detail on, over the timescales of sampling, what has (and could have) changed (e.g. land use, flow management, methods of sampling etc.,).

The reviewer asks a question about the filter pore size. Could you please cite the German study you mention, and/or provide a summary (few sentences) of how this was done? Is there any chance the filters being used changed through time for the longer timeseries study?

Their comment regarding L274 – I found this statement quite vague, so I would prefer to remove the jargon and instead explain the processes/factors in more detail.

Otherwise, please address these remaining comments that come from my reading of the original and revised manuscript: (Line numbers refer to the track changes manuscript provided in the response to reviews)

13 – Given the journal audience, and ambiguity of what LOI can be used for, it would be good to specify how the LOI is being used here. e.g. ". . . (LOI) of suspended matter at two stations along the rivers Moselle and Rhine to provide a proxy of the relative contributions of mineral load and organic matter".

18 – I It was a good idea to add something like this, which summarises the key process-level detail, but I find this new sentence very hard to follow. There is too much jargon, and it is quite vague. I think you are invoking both an increased supply of mineral load (erosion processes), but also a shift in organic matter source, from low mineral associated (i.e. high %LOI) aquatic biomass-derived organic matter at low flow, to mineral-associated organic matter (lower %LOI) eroded from the landscape at higher flow. If I'm correct, please summarise here.

20 – Somewhere in the abstract it would be useful to comment on how the SSC concentrations (mg/L) compare to global rivers – this would help to frame the studies findings, and perhaps show in which river systems the clearest comparisons could be found.

53 – to help set this in a wider literature, somewhere in here it would be worth specifying that this framework mostly applies to rivers with generally low turbidity and low suspended sediment concentrations (such as those found here, which are typically « 100 mg/L).

56 – I struggle a bit here – when you talk of low water flow velocities, the process of blooming phytoplankton and its accumulation basically needs zero flow velocity? Else it would be in motion downstream. What about primary producers in biofilms, or aquatic plants? What about leaf litter fall from riparian corridors?

61 – and at high flow runoff and erosion supply materials from outside the channel that swamp the within-river production?

129 – I think this is somewhat unfair given the large body of literature that examines particulate organic matter transport. There are numerous studies that examine POM (or POC) concentrations (% and mg/L), and specially examine it as a function of SSC and/or water discharge in catchments all over the world – New Zealand (Gomez et al., 2003, WRR), Taiwan (e.g. my work- Hilton et al., 2012, GBC), Swiss Alps (Smith et al., 2013, EPSL), USA (Hatten et al., 2010, Biogeochemistry), Peru (Clark et al., 2017, JGR), Guadeloupe (Lloret et al., 2011, Chemical Geology) to name but a few, none of which are referred to in this paper. It is also not just about acknowledging this literature, but also using it to help form broader conclusions. See comments below in the discussion

176 – could you please cite the work (mentioned in the replies that it is a German publication) and provide a brief outline here.

196 – why 'estimate'. Do you not 'measure' LOI?

199 – rephrase? – the whole sample is heated at 500oC, with an aim to combust the

**ESurfD**
[Figure]

organic fraction of the suspended matter.

200 – to help clarify further "ratio of the mass of organic matter (the mass loss on ignition) to the total suspended sediment mass (ranging from 0 to 1)."

215 – In here, please provide a brief overview of some of the challenges that surround LOI, in terms of different methods (temperatures, combustion lengths) and possibility that the weight loss does not only result from organic matter combustion (i.e. role of clay-bound water, carbonate decomposition etc.,). I think it would be useful to spell out that the LOI is used here as a proxy for the organic matter content – this is what you do, but a sentence stating that would be useful for the ESurf readership.

220 – This needs some more explanation. Were the LOI values used to estimate POC here? If so, please discuss with the caveats above.

410 – Remarkably, this is analogous to results when you examine soil and vegetation derived POC as a function of water discharge in mountain catchments (see Fig. 5 in Hilton et al., 2012, GBC). The reviewer 2 mentions this is to be expected, but I'm not sure too many studies have looked at this. Perhaps the authors can reflect on this - I think this could be worth some more discussion in the paper, with a view to explaining whether this feature should be more widely applicable.

500 – or viewed the other way, a lower dilution of this source (which contributes only a few mg/L) compared to higher flow, when it is swamped by mineral and catchment-derived OC? I'm not sure you can distinguish this explanation from the one given in the text.

540 – ok, but there is not much data to define this decrease on Figure 10.

547 – these ideas are strongly aligned with discussions from other papers on this topic (which are mentioned regarding line 129 above). In particular, Smith et al., 2013 EPSL, in section 5.3 (and check out their figures 2 and 5, for similarities to draw to this work) discussion very similar themes and mechanisms.

**ESurfD**

Interactive
comment

There is an opportunity here to draw parallels between these generally low turbidity rivers, and other work on catchments with higher sediment inputs. This could help generalise the findings. This discussion could be included in this section? Generally, this aspect of the discussion is well focused at present. But I wondered if there was an opportunity to take stock of how the process-understanding may make these features more common (or in fact, recognition that they may be specific to certain rivers?)

594 – The final line of the conclusions is not relevant to the findings here. Perhaps rephrase it, instead highlighting that more work is needed to see how generalised these findings could be, or something like that?

---

## Author Response (AR2)

**Reply to the Associate Editor (Robert Hilton) Decision from 28th April.**

By Thomas Hoffmann (on behalf of the co-authors) (May 10th, 2020)

Comments to Copernicus and the AE:

Dear Copernicus Team, dear Robert

First of all, I have to apologize for the delayed reply to the decision of the AE. Due to the Corona-Crisis, the closed Kindergarten and the order to work from home, I was not able to reply in time. We hope that the editors of ESurf will still consider our manuscript for publication.

We are very thankful to Robert Hilton for his valuable and constructive comments, which certainly increased the value of the manuscript. We considered every comment, changed the manuscript to implement the suggestions and gave a detailed reply in the following lines (indicated in green letters). We hope that the revised manuscript is not ready for publication.

Referee 2:
Their first comment about temporal changes – I didn't see much included on this point in the revised version. I think it's a fair point to make (indeed there are some discussions about temporal changes in the manuscript). It seems sensible to include some more detail on, over the timescales of sampling, what has (and could have) changed (e.g. land use, flow management, methods of sampling etc.,).

⟹ We added a full abstract at the beginning of the discussion reiterating the time scale of this study and make some statements about potential changes and impacts. We state that long-term changes of SSC (as observed at the gauging stations only effects the rating coefficient a, but not the rating exponent b, which is the focus of this study.

The reviewer asks a question about the filter pore size. Could you please cite the German study you mention, and/or provide a summary (few sentences) of how this was done? Is there any chance the filters being used changed through time for the longer timeseries study?

⟹ After discussions with the lab group and the co-authors we changed the argumentation ranging the pore size of the filters. We avoid to give a clear pore size, since pore sizes are not clearly defined for the coffee filters. However, we argue that a substantial fraction of the suspended clays is not recorded. We added some statements in the method chapter 2.2 and reiterated the uncertainty at the beginning of the discussion, by adding a new paragraph at the beginning of chapter 4.

Their comment regarding L274 – I found this statement quite vague, so I would prefer to remove the jargon and instead explain the processes/factors in more detail.

⟹ We rephrased this part to avoid the confusion, which were sated by reviewer II and the AE.

Otherwise, please address these remaining comments that come from my reading of the original and revised manuscript:

(note - line numbers refer to the track changes manuscript provided in the response to reviews)

13 – Given the journal audience, and ambiguity of what LOI can be used for, it would be good to specify how the LOI is being used here. e.g. "… (LOI) of suspended matter at two stations along the rivers Moselle and Rhine to provide a proxy of the relative contributions of mineral load and organic matter". → rephrased as suggested

18 – I It was a good idea to add something like this, which summarises the key process-level detail, but I find this new sentence very hard to follow. There is too much jargon, and it is quite vague. I think you are invoking both an increased supply of mineral load (erosion processes), but also a shift in organic matter source, from low mineral associated (i.e. high %LOI) aquatic biomass-derived organic matter at low flow, to mineral-associated organic matter (lower %LOI) eroded from the landscape at higher flow. If I'm correct, please summarise here. → yes, correct. We rephrased this sentence as suggested.

20 – Somewhere in the abstract it would be useful to comment on how the SSC concentrations (mg/L) compare to global rivers – this would help to frame the studies findings, and perhaps show in which river systems the clearest comparisons could be found. → we add that the concept refers to large (> 10 000 km2) and low turbid (SSC < 1000 mg/l) rivers, this should help to evaluate the value of this concept in a global context. Certainly, this SSC < 100 mg/l threshold is rather high, but SSC might rise up to several hundred mg/l during floods. Thus, this threshold is an upper boundary.

53 – to help set this in a wider literature, somewhere in here it would be worth specifying that this framework mostly applies to rivers with generally low turbidity and low suspended sediment concentrations (such as those found here, which are typically << 100 mg/L). → we added a similar state as in the abstract to the end of the introduction.

56 – I struggle a bit here – when you talk of low water flow velocities, the process of blooming phytoplankton and its accumulation basically needs zero flow velocity? Else it would be in motion downstream. What about primary producers in biofilms, or aquatic plants? What about leaf litter fall from riparian corridors?
   ⇒ We detailed the text in this paragraph. Basically, the background is: If phytoplankton grows at a certain growth rate (depending on light, temperature, nutrients), it can accumulate higher biomasses if it spends more time in a certain river stretch (i.e. at low flow velocities). Vice versa, if flow velocity is high, the growth rate cannot compensate the shorter water residence times and phytoplankton is washed out from the system.
   Primary producers in biofilms play a certain role as they also react on the factors mentioned above, but play a minor role in contributing to suspended load. They are therefore neglected in this discussion. Resuspended biofilm material would of course appear in the data, but its proportion is most probably extremely low as can be seen by lower proportion of organic matter at high discharges.
   Leaf litter contributes to the allochthonous suspended matter mentioned in the first

line of the paragraph, while the rest of the paragraph explicitly deals with autochthonously produced organic matter.

61 – and at high flow runoff and erosion supply materials from outside the channel that swamp the within-river production? → added

129 – I think this is somewhat unfair given the large body of literature that examines particulate organic matter transport. There are numerous studies that examine POM (or POC) concentrations (% and mg/L), and specially examine it as a function of SSC and/or water discharge in catchments all over the world – New Zealand (Gomez et al., 2003, WRR), Taiwan (e.g. my work- Hilton et al., 2012, GBC), Swiss Alps (Smith et al., 2013, EPSL), USA (Hatten et al., 2010, Biogeochemistry), Peru (Clark et al., 2017, JGR), Guadeloupe (Lloret et al., 2011, Chemical Geology) to name but a few, none of which are referred to in this paper. It is also not just about acknowledging this literature, but also using it to help form broader conclusions. See comments below in the discussion. → We are thankful for this suggestion and fully agree that this literature has been ignored. We added some of the references in the paragraph, which explains the expected relationship between POC and Q in the third paragraph of the introduction. We furthermore, included some of the results of this literature later in the discussion.

176 – could you please cite the work (mentioned in the replies that it is a German publication) and provide a brief outline here. → We added a refence (Hillebrand et al. 2015) and gave more information about the quality of the measured SSC based on coffee filters. We basically state that we underestimate SSC by approx. 20%, which is in the order of the clay content. We further argue that the clay content is not a function of discharge, and thus no discharge specific effects of the filter method are expected. In the discussion, we further argue that the same rating behavior is evidenced for the two LOI stations in Koblenz, where SSC is measured using standard glass fiber filters. These arguments should give sufficient information regarding the concern of the filter method.

196 – why 'estimate'. Do you not 'measure' LOI? → changed!

199 – rephrase? – the whole sample is heated at 500oC, with an aim to combust the organic fraction of the suspended matter. → done

200 – to help clarify further "ratio of the mass of organic matter (the mass loss on ignition) to the total suspended sediment mass (ranging from 0 to 1)." → done

215 – In here, please provide a brief overview of some of the challenges that surround LOI, in terms of different methods (temperatures, combustion lengths) and possibility that the weight loss does not only result from organic matter combustion (i.e. role of clay-bound water, carbonate decomposition etc.,). I think it would be useful to spell out that the LOI is used here as a proxy for the organic matter content – this is what you do, but a sentence stating that would be useful for the ESurf readership. → we added a sentence earlier in this section: "*Here we use the LOI as a proxy for the organic matter content of the suspended sediments, despite the challenges that are related to this method (i.e. different protocol regarding the temperature and combustion length result in various LOIs and combustion may*

*originate not only from organic matter but as well from clay-bound-water and carbonate decomposition)."*

220 – This needs some more explanation. Were the LOI values used to estimate POC here? If so, please discuss with the caveats above. → we added several lines at the end of this section on how we used the relationship of living biomass (as derived from Chla) and the organic fraction of SSC.

410 – Remarkably, this is analogous to results when you examine soil and vegetation derived POC as a function of water discharge in mountain catchments (see Fig. 5 in Hilton et al., 2012, GBC). The reviewer 2 mentions this is to be expected, but I'm not sure too many studies have looked at this. I think this could be worth some more discussion in the paper, with a view to explaining whether this feature should be more widely applicable. → thanks for this hint: we added some sentences to threshold hillslopes here and referred to Hilton et al. (2012)

500 – or viewed the other way, a lower dilution of this source (which contributes only a few mg/L) compared to higher flow, when it is swamped by mineral and catchment-derived OC? I'm not sure you can distinguish this explanation from the one given in the text. → correct, we rephrased this paragraph to add the aspect on dilution through catchment derived OC.

540 – ok, but there is not much data to define this decrease on Figure 10. → we added a not in the text on the rare observations of this decline.

547 – these ideas are strongly aligned with discussions from other papers on this topic (which are mentioned regarding line 129 above). In particular, Smith et al., 2013 EPSL, in section 5.3 (and check out their figures 2 and 5, for similarities to draw to this work) discussion very similar themes and mechanisms. → We related our finding to those of Smith et al and explained differences due to site specific sources and soil conditions.

There is an opportunity here to draw parallels between these generally low turbidity rivers, and other work on catchments with higher sediment inputs. This could help generalise the findings. This discussion could be included in this section? Generally, this aspect of the discussion is well focused at present. But I wondered if there was an opportunity to take stock of how the process-understanding may make these features more common (or in fact, recognition that they may be specific to certain rivers?) → at the end of the discussion, we argued that the findings are representative to large river systems with a similar human impact and indicated that more work is needed to test the application to other rives.

594 – The final line of the conclusions is not relevant to the findings here. Perhaps rephrase it, instead highlighting that more work is needed to see how generalised these findings could be, or something like that? → we rephrased the last sentence in the line of reasoning, as suggested by the AE.

---

## Author Response (AR3)

**Reply to the Associate Editor (Robert Hilton) Decision from 26th May.**

by Thomas Hoffmann (on behalf of the co-authors) (May 27th, 2020)

Comments to Copernicus and the AE:

Dear Copernicus Team, dear Robert,

thanks again for your comments on our MS. Your support is very much appreciated. Below you find a reply to each comment in your decision from 26th May.

10: "and water discharge (Q)" - a revision to clarify you're talking about water, and to define Q (used later in the abstract) → We added "(Q)" as suggested.

14: "power law rating curve" → We added "curve" in line 14.

31: "Transport of suspended" → We removed "the".

52: "production." → We added the point at the end of the sentence.

52: change "organic SSC" to "organic matter load"? → done!

79: change 'strong' to 'large'? → done!

95: This revised text is a bit awkward. Can I suggest a different way to say the same thing? Opening this paragraph with: "An alternative control on SSC - Q relationships may be the varied contribution of organic matter to river sediment loads. However, many studies that have investigated the composition and loading of organic matter are limited to a relatively narrow window (~year) of sample collection, and tend to focus on steep upload catchments (e.g. Goñi et al., 2014; Hilton et al., 2012; Smith et al., 2013). In contrast, studies with a large number of samples tend to focus on total suspended sediment without considering their mineral and organic components" → Thanks for your suggestion. Sounds much better. We exchanged the text as suggested.

165: I think the "estimate" should be "measure" → done

286: what does 'work progress' mean? Please rephrase. It would be useful to cite a paper here to back these claims up. → It means from 'work in progress'. We are working on an analysis of the temporal changes of SSC of the same dataset. From this analysis, we know that the exponent 'b' is not changing through time. Changes are limited to the 'a' coefficient. However, this work is not published yet, but will be submitted very soon.

327: I think it helps broaden the findings. Note – is POC defined anywhere. Perhaps just replace "POC" with "particulate organic carbon concentration and discharge" → We changed the text to "particulate organic carbon (POC)" since we use the acronym later on.

[revised manuscript text omitted]

---

## Author Response (AR4)

**Reply to the Editor (Josh West) Decision from July 6th.**

By Thomas Hoffmann (on behalf of the co-authors) (July 8th, 2020)

Dear Copernicus Team, dear Josh

Thanks very much for your detailed edits on our manuscript. All your comments were considered and the text changed as suggested.

We added the reference of Morin et al (2018).

Figure 1: We added an ETRS grid to the figure and located the stations in Koblenz (only one dot is visible as they overlap due to their close distance). Federal States were removed in an earlier version of the figure but the legend entry was still there. This is removed now.

Figure 3: we explained the meaning of Q_crv, Q_rge and Q_bin in the Figure caption.

Figure 8: we increased the size of the labels and the legend text.

Table 1: GM is explained in the caption of the table, similar to avg and med. We avoid to have a legend of the table. We explained the location and monitoring period, in the case if only one year is given.

Table 2: We explained the capital delta in the table caption, changed all "," to "." and checked the alignment of the text in the rows.

Kind regards
Thomas Hoffmann